# Detecting Discrepancies Between AI-Generated and Natural Images Using Uncertainty

## Abstract

In this work, we propose a novel approach for detecting AI-generated images by leveraging predictive uncertainty to mitigate misuse and associated risks. The motivation arises from the fundamental assumption regarding the distributional discrepancy between natural and AI-generated images. **The feasibility of distinguishing natural images from AI-generated ones is grounded in the distribution discrepancy between them**. Predictive uncertainty offers an effective approach for capturing distribution shifts, thereby providing insights into detecting AI-generated images. Namely, as the distribution shift between training and testing data increases, model performance typically degrades, often accompanied by increased predictive uncertainty. Therefore, we propose to employ predictive uncertainty to reflect the discrepancies between AI-generated and natural images. In this context, the challenge lies in ensuring that the model has been trained over sufficient natural images to avoid the risk of determining the distribution of natural images as that of generated images. We propose to leverage large-scale pre-trained models to calculate the uncertainty as the score for detecting AI-generated images. Inspired by MC Dropout, we perturb pre-trained models and find that the uncertainty can be captured by perturbing the weights of pre-trained models. This leads to a simple yet effective method for detecting AI-generated images using large-scale vision models: images that induce high uncertainty are identified as AI-generated. Comprehensive experiments across multiple benchmarks demonstrate the effectiveness of our method.

## 1 Introduction

Recent advancements in generative models have revolutionized image generation, enabling the production of highly realistic images (Midjourney; Wukong; Rombach et al., 2022). Despite the remarkable capabilities of these models, they pose significant challenges, particularly the rise of deepfakes and manipulated content. The high degree of realism achievable by such technologies prompts urgent discussions about their potential misuse, especially in sensitive domains such as politics and economics. In response to these critical concerns, a variety of methodologies for detecting generated images have emerged. A prevalent strategy treats this detection task as a binary classification problem, necessitating the collection of extensive datasets comprising both natural and AI-generated images to train classifiers (Wang et al., 2020).

While existing detection methods have demonstrated notable successes, they typically encounter challenges in generalizing to images produced by previously unseen generative models Wang et al. (2023a). One promising avenue to enhance the robustness of detection capabilities involves constructing more extensive training datasets by accumulating a diverse array of natural and synthetic images. However, these attempts are often computationally intensive, requiring substantial datasets for effective binary classification. Additionally, maintaining robust detection necessitates continually acquiring images generated by the latest models. And when the latest generative models are not open-sourced, acquiring a large number of generated

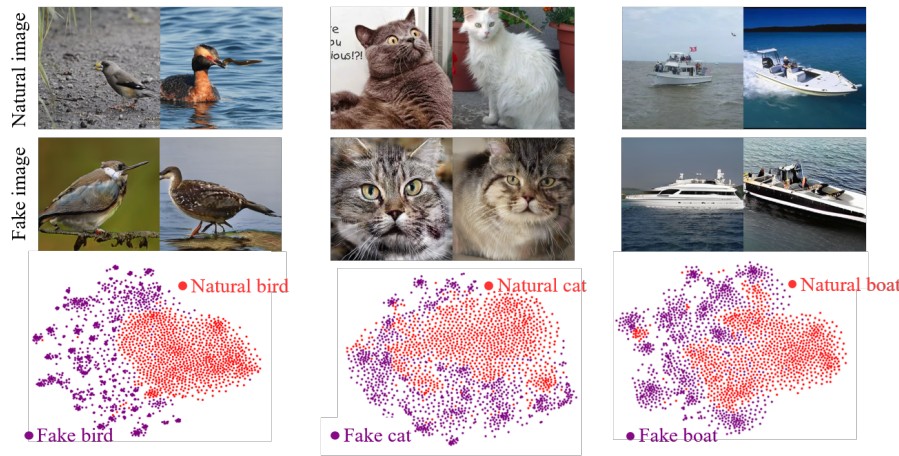

Figure 1: Large-scale fundamental models trained on a large number of real images are capable of distinguishing between real and generated images.

images to train classifiers is challenging. This highlights the urgent need for a novel framework to detect AI-generated images without requiring AI-generated images.

A recent work (Tan et al., 2024) shows that features extracted by ViT of CLIP (Radford et al., 2021) can be employed to separate natural and AI-generated images, motivating an effective approach to detecting images by training a binary classifier in the feature space of CLIP. This provides a promising direction to explore the possibility that large-scale foundational models already have the ability to capture the subtle differences between real images and AI-generated images. This may be similar to the emergence phenomenon in large language models (Wei et al., 2022), even if these models are not designed to distinguish between real and generated images. This phenomenon may arise from the training data of large vision models. In particular, since training models with AI-generated data usually leads to model collapse (Shumailov et al., 2024), current large vision models are mainly trained over natural data, making these models biased between natural and AI-generated data. Even though natural and AI-generated images appear extremely semantically similar to humans, such "biased" large models can capture the difference between natural and generated images.

The distribution discrepancy in features between natural and generated images motivates us to revisit the strategy of detecting AI-generated images. Specifically, the ability of humans to distinguish between natural and generated images relies on the existence of the discrepancy between them. Moreover, humans are not trained to distinguish natural and generated images. To verify this intuition, we visualize the features of natural and generated images using a large-scale visual fundamental model (DINOv2), following previous work (Tan et al., 2024). As shown in Figure 1, even for images sampled from the same class, there are large distributional discrepancies in the feature space of DINOv2.

Building on this foundation, we propose to leverage predictive uncertainty as the score for distinguishing between natural and AI-generated images. This is because predictive uncertainty offers an effective approach for capturing distribution shifts. In particular, recent studies (Snoek et al., 2019; Schwaiger et al., 2020) indicate that models tend to show increased uncertainty for out-of-distribution (OOD) samples. Hence, for large vision models trained only on natural images, we can treat the natural images as in-distribution samples and the generated images as OOD samples. The challenge comes from efficiently obtaining the uncertainty of the model on the test samples. Classical approaches include Monte-Carlo Dropout (MC-Dropout) (Gal & Ghahramani, 2016) and Deep Ensembles (Lakshminarayanan et al., 2017). However, in our attempts, MC-Dropout obtains sub-optimal results (in Table 5), and it is not practical to train multiple large models

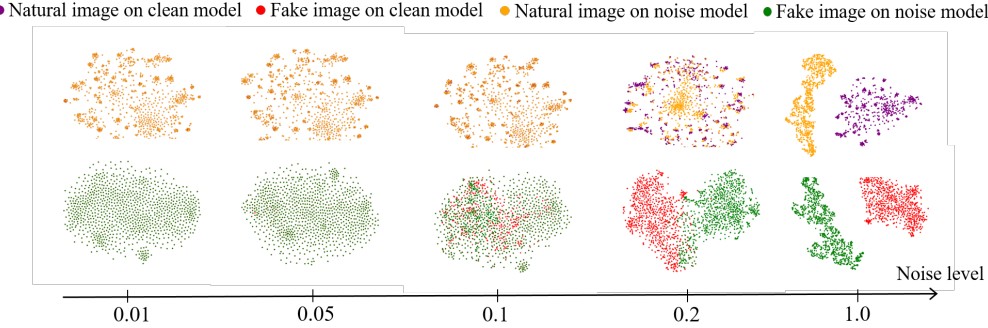

Figure 2: Natural and generated images are differently sensitive to perturbations in the model weights. When a moderate perturbation is applied (0.1), the natural image has essentially the same features on the model before and after the perturbation, but the generated image already shows a large difference.

independently for ensemble. Instead, we find that this uncertainty can be well captured by perturbing the weights of the models. As shown in Figure 2, when a moderate level of perturbation is applied, the real image has consistent features on the model before and after the perturbation, but the generated image has large differences in features on the model before and after the perturbation.

In this paper, we propose a novel method for AI-generated image detection by weight perturbation (WePe). Our hypothesis is that the model has greater uncertainty in predicting the OOD sample compared to the ID sample, and that this uncertainty can be expressed through sensitivity to weight perturbations. For a large model trained on a large number of real images, the real images can be considered ID samples, while the generated images are considered OOD samples. Thus, the sensitivity of the samples to the weight perturbation of the large model can be an important indicator to determine whether the generative models generate the samples. Despite its simplicity, WePe achieves state-of-the-art performance on various benchmarks.

We summarize our main contributions as follows:

- We provide a new perspective to detect AI-generated images by calculating predictive uncertainty. This is built upon an intuitive assumption that the natural and generated images differ in distribution, making it possible to employ uncertainty to represent the distribution discrepancy.

- We propose to leverage large vision models to calculate the predictive uncertainty. The intuition is that large vision models are merely trained on natural images, making it possible to exhibit different uncertainties about natural and generated images. We capture this uncertainty by weight perturbation to effectively detect images (Eq. 3).

- Comprehensive experiments on multiple benchmarks demonstrate that the proposed method outperforms previous methods, including training-based methods, achieving state-of-the-art performance.

## 2 RELATED WORKS

**AI-Generated images detection.** Recent advancements in generative models, such as those by (Brock et al., 2019; Ho et al., 2020), have led to the creation of highly realistic images, highlighting the urgent need for effective algorithms to distinguish between natural and generated images. Prior research, including works by (Frank et al., 2020; Marra et al., 2018), primarily focuses on developing specialized binary classification neural networks to differentiate between natural and generated images. Notably, CNNspot (Wang et al., 2020) demonstrates that a standard image classifier trained on ProGAN can generalize across various architectures when combined with specific data augmentation techniques. F3Net (Qian et al., 2020) distinguishes

real face images from fake face images with the help of frequency statistical differences. NPR (Tan et al., 2024) introduces the concept of neighboring pixel relationships to capture differences between natural and generated images. Although these methods show superior performance on generators in the training set, they often do not generalize well to unknown generators. In addition to this, training-based methods are susceptible to small perturbations in the image. For this reason, recently, some training-free methods have been proposed. AEROBLADE (Ricker et al., 2024) calculates the reconstruction error with the help of the autoencoder used in latent diffusion models (Rombach et al., 2022). RIGID (He et al., 2024) finds that real images are more robust to small noise perturbations than generated images in the representation space of the vision foundation models and exploits this property for detection. However, these methods usually make overly strong assumptions about natural or generated images, leading to insufficient generalization. In our paper, we propose a training-free detection method through uncertainty analysis. Based on the widespread phenomenon that generated images have greater uncertainty than real images on models trained with real images, our method achieves robust detection performance.

**Uncertainty estimation.** Uncertainty estimation in machine learning has seen significant advancements in recent years. (Gal & Ghahramani, 2016) introduces Monte Carlo Dropout (MC Dropout), which uses dropout at inference to estimate uncertainty from the variance of multiple predictions. (Lakshminarayanan et al., 2017) develops deep ensembles, demonstrating improved uncertainty estimates through training multiple model indepently with different initializations. Recent work by (Snoek et al., 2019) analyzes the calibration of uncertainty in deep learning models, highlighting the importance of reliable uncertainty measures. Additionally, (Guo et al., 2017) explore the use of temperature scaling to enhance the calibration of model predictions. In our paper, We measure uncertainty by perturbing the model's weights.

**Random weight perturbations.** Weight perturbation, i.e. adding noise to network weights, has been extensively studied. Many methods (Khan et al., 2018; Wu et al., 2020) perform training regularization by adding noise to the weights of the neural network during training to improve the generalization of the network. Weight perturbation has also been applied in adversarial attacks to study the robustness of the network (He et al., 2019; Garg et al., 2020). In addition to this, some works also study the sensitivity of neural network to weights perturbation. (Cheney et al., 2017) shows that convolutional networks are surprisingly robust to a number of internal perturbations in the higher convolutional layers but the bottom convolutional layers are much more fragile. (Weng et al., 2020) proposes an efficient approach to compute a certified robustness bound of weight perturbations. In our work, we find neural networks trained on natural images exhibit different robustness to weight perturbations for natural and generated images. Based on this property, we propose an efficient algorithm to distinguish between natural and generated images.

## 3 METHOD

### 3.1 MOTIVATION

Our method is built upon a foundational assumption that natural and generated images have different distributions. This is a reasonable assumption; otherwise, we cannot distinguish between natural and generated images. Fortunately, this assumption is consistent with previous work (Tan et al., 2024) and our empirical observations, as shown in Figure 1.

Hence, for the large models trained merely on natural images, we can regard natural images as in-distribution (ID) data while generated images as out-of-distribution (OOD) data. This distribution discrepancy can be reflected by the widely used predictive uncertainty, since neural networks typically exhibit higher uncertainty for OOD samples (Snoek et al., 2019; Schwaiger et al., 2020). This leads to a simple yet novel approach to determine whether a test image is generated by AI models when we can calculate its uncertainty on a pre-trained large vision model. In the following, we first give a complete introduction to our method WePe, then describe on our reasons for choosing the method, and finally, we explore why WePe works.

## 3.2 UNCERTAINTY DEFINITION

Classical methods of uncertainty estimation, such as Deep Ensembles and MC Dropout, can simply be viewed as using the variance of the results of multiple predictions as an estimation of uncertainty $u(\mathbf{x})$:

$$\mu(\mathbf{x}) = \frac{1}{n} \sum_{t=1}^{n} \hat{y}_t(\mathbf{x}), \quad u(\mathbf{x}) = \sigma^2 = \frac{1}{n} \sum_{t=1}^{n} \left( \hat{y}_t(\mathbf{x}) - \mu(\mathbf{x}) \right)^2, \tag{1}$$

where, $\hat{y}_t$ denotes the $t$-th prediction. The multiple predictions of Deep Ensembles come from multiple independently trained neural networks, while the multiple predictions of MC Dropout come from the use of dropout during inference, which can be regarded as multiple prediction using neural networks with different structures.

## 3.3 UNCERTAINTY CALCULATION

The predictive uncertainty is typically calculated as the variance of predictions obtained with certain perturbations. In this work, we simply leverage $\theta$ as the features or parameters before perturbation. Specifically, the predictive uncertainty $u(\mathbf{x})$ can be calculated by,

$$u(\mathbf{x}) = \frac{1}{n} \sum_{k=1}^{n} [f(\mathbf{x}; \theta_k)^\top f(\mathbf{x}; \theta_t) - \sum_{j=1}^{n} \frac{f(\mathbf{x}; \theta_j)^\top f(\mathbf{x}; \theta_t)}{n}]^2, \tag{2}$$

where $n$ is the number of samples, $f(\mathbf{x}; \theta_k)$ denotes the L2-normalized features of an input image $\mathbf{x}$ when inferring with the parameter $\theta_k$, and $\theta_t$ stands for the teacher model used in DINOv2.

However, we cannot access the teacher model $\theta_t$, making it challenging to calculate the uncertainty. Moreover, even if it is available, introducing two models for calculation leads to low computation efficiency. Fortunately, we can calculate an upper bound of $u(\mathbf{x})$. This can be formalized by,

$$u(\mathbf{x}) \leq \frac{1}{n} \sum_{k=1}^{n} \left\| f(\mathbf{x}; \theta_k) - \frac{1}{n} \sum_{j}^{n} f(\mathbf{x}; \theta_j) \right\|^2 \|f(\mathbf{x}; \theta_t)\|^2 = 2 - \frac{2}{n} \sum_{k=1}^{n} f(\mathbf{x}; \theta_k)^\top f(\mathbf{x}; \theta), \tag{3}$$

where $\theta$ denotes the parameter before injecting perturbation, and we leverage an unbiased assumption that the expectation $\mathbb{E}_{\theta_j} f(\mathbf{x}; \theta_j)$ approaches the feature $f(\mathbf{x}; \theta)$ extracted by the non-perturbed parameter $\theta$. Eq. 3 provides a simple approach to calculate the uncertainty without needing a teacher model used in the training phase of DINOv2. The insight of Eq. 3 is intuitive. Specifically, if an image $\mathbf{x}$ causes a high feature similarity between the original and perturbed parameter, the image leads to a low uncertainty and is more likely to be a natural image.

## 3.4 AN OVERVIEW OF WEPE

As discussed above, the proposed WePe is based on a large model pre-trained on a large number of natural images. In this work, we chose DINOv2 (Oquab et al., 2024), a large model trained with contrastive learning on image data. In order to capture the uncertainty of the model on the test images, we extract image features using the original model and the model after adding noise to the parameters respectively. The similarity between the pre-perturbation and post-perturbation feature vectors is quantified using a suitable distance metric, such as cosine similarity. Images exhibiting high similarity are classified as real, while those with low similarity are identified as generated. This method not only capitalizes on the characteristics of the DINOv2 model but also provides a robust framework for distinguishing between real and generated images based on their feature stability under model perturbation.

## 3.5 DISCUSSIONS

**Why choose DINOv2?** In addition to DINOv2, CLIP is a commonly used model. However, our experiments show that CLIP performs sub-optimally compared to DINOv2 (see Table 4). We believe this difference stems from their training strategies. Unlike DINOv2, CLIP is a multimodal model that combines image and textual features from captions for contrastive learning, which may lead it to focus on broader semantic features rather than fine details. In contrast, DINOv2 emphasizes contrastive learning solely on images, allowing it to better capture subtle differences between natural and generated images. Therefore, we use DINOv2 in our experiments.

**Why choose weight perturbation?** Common methods for measuring uncertainty include MC Dropout and Deep Ensembles. MC Dropout involves keeping dropout active during testing and performing multiple forward passes on the inputs to generate outputs with different network structures. The variability among these outputs serves as an estimate of the model's uncertainty regarding the input data. However, since DINOv2 does not utilize dropout during training, MC Dropout may not yield optimal results (see Table 5). Deep Ensembles, on the other hand, trains multiple networks independently and uses the differences in their outputs on test samples to assess uncertainty. However, training multiple DINOv2-level models is impractical. Therefore, in our study, we choose to perturb the model parameters and assess the differences in outputs from the original and perturbed models to estimate uncertainty for the test images.

**Why does weight perturbation work?** Incorporating weight perturbations during testing can effectively simulate Bayesian inference, thereby capturing the inherent uncertainty in neural networks. From a Bayesian perspective, the weights of a neural network are not fixed but rather distributions reflecting the range of plausible values given the data. By introducing noise into the network weights at test time, we mimic the process of drawing samples from a posterior distribution over weights, a core concept in Bayesian inference. This technique enables the model to generate diverse predictions, reflecting its uncertainty, particularly when encountering out-of-distribution samples. This approach is akin to Bayesian neural networks, where weight uncertainty is explicitly modeled. For example, (Blundell et al., 2015) proposed using variational methods to approximate weight distributions in Bayesian neural networks, allowing for uncertainty quantification through weight perturbations. Similarly, (Gal & Ghahramani, 2016) demonstrated that introducing dropout at test time serves as a Bayesian approximation, with the added noise acting as a proxy for weight sampling, thus allowing for reliable uncertainty estimation. In this paper, we capture uncertainty by adding noise directly to the model parameters to distinguish between natural and generated images. However, trying to theoretically analyze the different sensitivities of natural and generated images to model weights in the representation space of a large model is a difficult task. Therefore, we only draw this conclusion through various empirical observations in this paper, and do not directly prove this theoretically. The theoretical proof will be left to our future work.

## 4 EXPERIMENTS

### 4.1 EXPERIMENT SETUP

**Datasets.** Following previous work (He et al., 2024), we evaluate the performance of WePe on ImageNet (Deng et al., 2009), LSUN-BEDROOM (Yu et al., 2015) and GenImage (Zhu et al., 2023). For ImageNet and LSUN-BEDROOM, the generated images are provided by (Stein et al., 2023). For ImageNet, the generative models include ADM (Dhariwal & Nichol, 2021), ADM-G, LDM (Rombach et al., 2022), DiT-XL2 (Peebles & Xie, 2023), BigGAN (Brock et al., 2019), GigaGAN (Kang et al., 2023), StyleGAN (Karras et al., 2019), RQ-Transformer (Lee et al., 2022), and MaskGIT (Chang et al., 2022). For LSUN-BEDROOM, the generative models include ADM, DDPM (Ho et al., 2020), iDDPM (Nichol & Dhariwal, 2021), Diffusion Projected GAN (Wang et al., 2023b), Projected GAN (Wang et al., 2023b), StyleGAN (Karras et al., 2019) and Unleasing Transformer (Bond-Taylor et al., 2022). GenImage primar-

ily employs the Diffusion model for image generation. The fake images are generated by Stable Diffusion V1.4 (Rombach et al., 2022), Stable Diffusion V1.5 (Rombach et al., 2022), GLIDE, VQDM (Gu et al., 2022), Wukong (Wukong), BigGAN, ADM, and Midjourney (Midjourney). It contains 1,331,167 real and 1,350,000 generated images.

**Evaluation metrics.** Following RIGID, we mainly use the following metrics: (1) the average precision (AP), (2) the area under the receiver operating characteristic curve (AUROC). On GenImage, since the dataset is too large and re-implementing all baselines is time-consuming, we obtain the results directly from the corresponding papers and report the classification accuracy (ACC).

**Baselines.** Following RIGID, we take both training-free methods and training methods as baselines. For training-free methods, we take RIGID (He et al., 2024) and AEROBLADE (Ricker et al., 2024) as baselines. For training methods, we take DIRE (Wang et al., 2023a), CNNspot (Wang et al., 2020), Ojha (Ojha et al., 2023) and NPR (Tan et al., 2024) as baselines. Besides, on GenImage, we also report the result of Frank (Frank et al., 2020), Durall (Durall et al., 2020), Patchfor (Chai et al., 2020), F3Net (Qian et al., 2020), SelfBland (Shiohara & Yamasaki, 2022), GANDetection (Mandelli et al., 2022), LGrad (Tan et al., 2023), ResNet-50 (He et al., 2016), DeiT-S (Touvron et al., 2021), Swin-T (Liu et al., 2021), Spec (Zhang et al., 2019), GramNet (Liu et al., 2020). For these baselines, we get the results directly in the corresponding papers without reimplementing them.

**Experiment details.** To balance detection performance and efficiency, we use DINOv2 ViT-L/14. We report the average results under five different random seeds and report the variance in Figure 8. In our experiments we find that perturbing the high layers may lead to a large corruption in the features of the real images, resulting in sub-optimal results. Therefore, We do not perturb the high-level parameters. In DINOv2 ViT-L/14, the model has 24 transformer blocks, and we only perturb the parameters of the first 19 blocks with Gaussian perturbations of zero mean. The variance of the added Gaussian noise is proportional to the mean value of the parameters in each block, with the ratio set to 0.1. Ablation experiments show that our method is robust to the blocks chosen for perturbation and the level of noise.

## 4.2 RESULTS

**Comparison with other baselines.** We conduct full comparative experiments on the three benchmarks mentioned. As shown in Table 1, 2 and 3, WePe achieves the best detection performance on ImageNet, LSUN-BEDROOM and GenImage without the need for training. It is worth noting that on the large-scale GenImage benchmark, the training-based method, despite having perfect performance on the generators used during training, performs poorly on many generators not seen during training. This illustrates the extreme dependence of the performance of training-based methods on the diversity of the training set. In contrast, our method does not require training, performs well on a wide variety of generators, and outperforms the SOTA training method by 3.36% on average. On generators that have not been seen during training, such as VQDM, many training methods exhibit random prediction results, whereas WePe achieves superior detection performance. To further illustrate the effectiveness of our method, we count the difference in feature similarity between real and fake images on the pre- and post-perturbation models. As shown in Figure 3, the small perturbation of the model has less effect on the real images than on generated images, resulting in higher feature similarity before and after the perturbation. The discrepancy effectively distinguishes the real image from the generated image.

## 4.3 ABLATION STUDY

In this section, we perform ablation experiments. Unless otherwise stated, experiments are conducted on ImageNet benchmark.

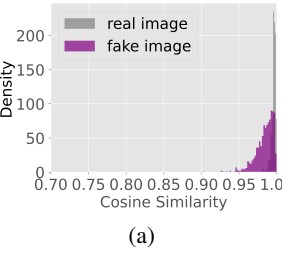 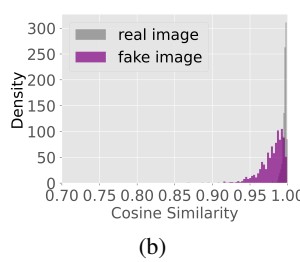 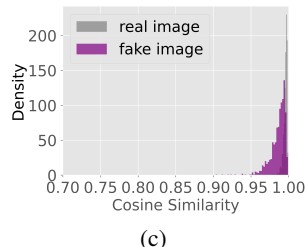

(a)  (b)  (c)

Figure 3: Comparison of cosine similarity between original and transformed images. The generated images are from: (a) ADM, (b) BigGAN, and (c) DDPM.

Table 1: AI-generated image detection performance on ImageNet. Values are percentages. **Bold** numbers are superior results and the underlined italicized values are the second-best performance. A higher value indicates better performance.

| Methods | ADM | | ADMG | | LDM | | DiT | | BigGAN | | GigaGAN | | StyleGAN XL | | RQ-Transformer | | Mask GIT | | Average | |
|---|---|---|---|---|---|---|---|---|---|---|---|---|---|---|---|---|---|---|---|---|
| | AUROC | AP | AUROC | AP | AUROC | AP | AUROC | AP | AUROC | AP | AUROC | AP | AUROC | AP | AUROC | AP | AUROC | AP | AUROC | AP |
| | | | | | | | | | Training Methods | | | | | | | | | | | |
| CNNspot | 62.25 | 63.13 | 63.28 | 62.27 | 63.16 | 64.81 | 62.85 | 61.16 | 85.71 | 84.93 | 74.85 | 71.45 | 68.41 | 68.67 | 61.83 | 62.91 | 60.98 | 61.69 | 67.04 | 66.78 |
| Ojha | 83.37 | 82.95 | 79.60 | 78.15 | 80.35 | 79.71 | 82.93 | 81.72 | 93.07 | 92.77 | 87.45 | 84.88 | 85.36 | 83.15 | 85.19 | 84.22 | 90.82 | 90.71 | 85.35 | 84.25 |
| DIRE | 51.82 | 50.29 | 53.14 | 52.96 | 52.83 | 51.84 | 54.67 | 55.10 | 51.62 | 50.83 | 50.70 | 50.27 | 50.95 | 51.36 | 55.95 | 54.83 | 52.58 | 52.10 | 52.70 | 52.18 |
| NPR | 85.68 | 80.86 | 84.34 | 79.79 | 91.98 | 86.96 | 86.15 | 81.26 | 89.73 | 84.46 | 82.21 | 78.20 | 84.13 | 78.73 | 80.21 | 73.21 | 89.61 | 84.15 | 86.00 | 80.84 |
| | | | | | | | | | Training-free Methods | | | | | | | | | | | |
| AEROBLADA | 55.61 | 54.26 | 61.57 | 56.58 | 62.67 | 60.93 | 85.88 | 87.71 | 44.36 | 45.66 | 47.39 | 48.14 | 47.28 | 48.54 | 67.05 | 67.69 | 48.05 | 48.75 | 57.87 | 57.85 |
| RIGID | 87.16 | 85.08 | 80.09 | 77.07 | 72.43 | 69.30 | 70.40 | 65.94 | 90.08 | 89.26 | 86.39 | 84.11 | 86.32 | 85.44 | 90.06 | 88.74 | 89.30 | 89.25 | 83.58 | 81.58 |
| WePe | 89.79 | 87.32 | 83.20 | 78.80 | 78.47 | 73.50 | 77.13 | 71.21 | 94.24 | 93.64 | 92.15 | 90.29 | 93.86 | 92.86 | 93.50 | 91.47 | 89.55 | 86.25 | 87.99 | 85.04 |

**Robustness to Perturbations.** Robustness to various perturbations is a critical metric for detecting generated images. In real-world scenarios, images frequently undergo perturbations that can impact detection performance. Following RIGID, we assess the robustness of detectors against three types of perturbations: JPEG compression (with quality parameter $q$)), Gaussian blur (with standard deviation $\sigma$), and Gaussian noise (with standard deviation $\sigma$). As illustrated in Figure 6, training-free methods generally exhibit superior robustness compared to training-based methods, with our method achieving the best overall performance.

**The impact of the degree of perturbation.** As shown in Figure 4, we explore the effect of the degree of perturbation on the performance of WePe. It can be seen that WePe is quite robust to the level of perturbation noise. It is only when the noise is very large or very small that it leads to a degradation in performance. When the noise level is small, the features obtained before and after the model perturbation are extremely similar, while when the noise level is very large, the features obtained before and after the model perturbation are extremely dissimilar, and these two cases will result in the inability to effectively differentiate between real and generated images.

**Selecting which layers' parameters to perturb?** As shown in Figure 5, we explore the choice of which layers' parameters to perturb would achieve good performance. The horizontal coordinates in the graph indicate that the first $k$ blocks are perturbed, not the $kth$ block. The experimental results exhibit that our method obtains good performance when the parameters of the first 9 to the first 20 blocks are chosen to be perturbed. This demonstrates the robustness of our method. In practice, we can select the layers to be perturbed by a small set of real and generated images. And when the generated images are not available, we can also use the probe to determine which layers are perturbed using only the real image. We describe our method in Appendix A.15.

**The effect of models.** In our experiments, we mainly used DINOv2 ViT-L/14 to extract features. We further explore the effect of using other models of DINOv2, including ViT-S/14, ViT-B/14, and ViT-g/14. In

Table 2: AI-generated image detection performance on LSUN-BEDROOM.

| Methods | ADM | | DDPM | | iDDPM | | Diffusion GAN | | Projected GAN | | StyleGAN | | Unleashing Transformer | | Average | |
|---|---|---|---|---|---|---|---|---|---|---|---|---|---|---|---|---|
| | AUROC | AP | AUROC | AP | AUROC | AP | AUROC | AP | AUROC | AP | AUROC | AP | AUROC | AP | AUROC | AP |
| CNNspot | 64.83 | 64.24 | 79.04 | 80.58 | 76.95 | 76.28 | 88.45 | 87.19 | 90.80 | 89.94 | **95.17** | **94.94** | 93.42 | 93.11 | 84.09 | 83.75 |
| Ojha | 71.26 | 70.95 | 79.26 | 78.27 | 74.80 | 73.46 | 84.56 | 82.91 | 82.00 | 78.42 | 81.22 | 78.08 | 83.58 | 83.48 | 79.53 | 77.94 |
| DIRE | 57.19 | 56.85 | 61.91 | 61.35 | 59.82 | 58.29 | 53.18 | 53.48 | 55.35 | 54.93 | 57.66 | 56.90 | 67.92 | 68.33 | 59.00 | 58.59 |
| NPR | **75.43** | **72.60** | **91.42** | **90.89** | **89.49** | **88.25** | 76.17 | 74.19 | 75.07 | 74.59 | 68.82 | 63.53 | 84.39 | 83.67 | 80.11 | 78.25 |
| AEROBLADA | 57.05 | 58.37 | 61.57 | 61.49 | 59.82 | 61.06 | 47.12 | 48.25 | 45.98 | 46.15 | 45.63 | 47.06 | 59.71 | 57.34 | 53.85 | 54.25 |
| RIGID | 71.90 | 72.29 | 88.31 | 88.55 | 84.02 | 84.80 | 91.42 | 91.90 | 92.12 | 92.54 | 77.29 | 74.96 | 91.37 | 91.39 | 85.20 | 85.20 |
| WePe | 73.85 | 70.21 | 88.84 | 87.14 | 86.23 | 83.82 | **94.16** | **93.52** | **95.34** | **95.18** | 83.50 | 80.66 | **94.18** | **93.45** | **88.01** | **86.28** |

Table 3: AI-generated image detection performance on GenImage. Except for WePe and RIGID, all methods require training on SD V1.4. The results of the baseline method are all from GenImage

| | Models | | | | | | | |
|---|---|---|---|---|---|---|---|---|
| Methods | Midjourney | SD V1.5 | ADM | GLIDE | Wukong | VQDM | BigGAN | Average |
| ResNet-50 | 54.90 | 99.70 | 53.50 | 61.90 | 98.20 | 56.60 | 52.00 | 68.11 |
| DeiT-S | 55.60 | 99.80 | 49.80 | 58.10 | 98.90 | 56.90 | 53.50 | 67.51 |
| Swin-T | 62.10 | 99.80 | 49.80 | 67.60 | 99.10 | 62.30 | 57.60 | 71.19 |
| CNNspot | 52.80 | 95.90 | 50.10 | 39.80 | 78.60 | 53.40 | 46.80 | 58.63 |
| Spec | 52.00 | 99.20 | 49.70 | 49.80 | 94.80 | 55.60 | 49.80 | 64.41 |
| F3Net | 50.10 | **99.90** | 49.90 | 50.00 | **99.90** | 49.90 | 49.90 | 64.22 |
| GramNet | 54.20 | 99.10 | 50.30 | 54.60 | 98.90 | 50.80 | 51.70 | 65.66 |
| DIRE | 60.20 | 99.80 | 50.90 | 55.00 | 99.20 | 50.10 | 50.20 | 66.49 |
| Ojha | 73.20 | 84.00 | 55.20 | 76.90 | 75.60 | 56.90 | 80.30 | 71.73 |
| LaRE | 66.40 | 87.10 | 66.70 | 81.30 | 85.50 | 84.40 | 74.00 | 77.91 |
| RIGID | **81.54** | 68.72 | 72.35 | **84.15** | 68.57 | 78.98 | **93.02** | 78.19 |
| WePe | 79.17 | 75.57 | **76.07** | 79.20 | 79.00 | **90.60** | 89.27 | **81.27** |

addition to this, we conduct experiments on the CLIP:ViT-L/14. As shown in Table 4, the performance on CLIP is not as good as on DINOv2. We hypothesize that the difference comes from the training approach of these models. CLIP learns features using image captions as supervision, which may make the features more focused on semantic information, whereas DINOv2 learns features only from images, which makes it more focused on the images themselves, and thus better able to capture subtle differences in real and generated images. When using DINOv2, our methods performs poorly when the capacity of the model is small. This may be due to the fact that larger models can better capture the differences between real and fake images. In our main experiments, to balance detection efficiency and detection performance, we use DINOv2 ViT-L/14.

**The effect of perturbation type.** In our experiments, we perturb the model parameters by adding Gaussian noise. We further explore other ways of perturbation, such as adding uniform noise or Laplace noise to the weight. In addition to this, we also explore the effect of MC Dropout, i.e., using dropout during inference. As shown in Table 5, all three weight perturbation methods achieve good performance, and outperform MC Dropout.

Table 4: The effect of models.

| model | AUROC | AP |
|---|---|---|
| DINOv2: ViT-S/14 | 72.83 | 71.63 |
| DINOv2: ViT-B/14 | 81.82 | 80.64 |
| DINOv2: ViT-L/14 | 87.99 | 85.04 |
| DINOv2: ViT-g/14 | 84.92 | 81.83 |
| CLIP: ViT-L/14 | 77.89 | 77.90 |

Table 5: The effect of type of perturbation.

| noise | AUROC | AP |
|---|---|---|
| Gaussian noise | 87.99 | 85.04 |
| Uniform noise | 89.06 | 86.32 |
| Laplace noise | 87.13 | 84.22 |
| MC Dropout | 81.63 | 79.71 |

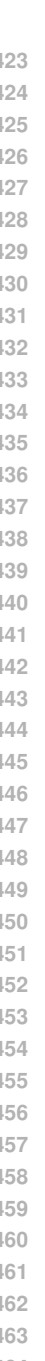
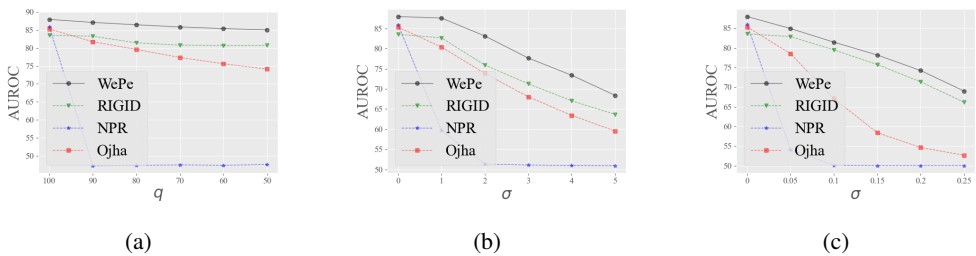

Figure 4: Performance varies with variance

Figure 5: The effects of disturbed blocks.

| (a) | (b) | (c) |

Figure 6: Illustration of detection performance varying with perturbation intensity under different degradation mechanisms, (a): JPEG compression, (b): Gaussian blur, and (c): Gaussian noise.

## 5 LIMITATION

The proposed weight perturbation provides a simple and effective method for detecting generated images, yet we have not theoretically justified the widespread use of the method due to the inclusion of a variety of strong prior assumptions, such as the assumption about treating generated samples as OOD data. Therefore, our future work will focus on establishing the theoretical foundations of our method.

## 6 CONCLUSION

In this work, to effectively address the challenges of detecting AI-generated images, we propose a novel approach that leverages predictive uncertainty as a key metric. Our findings reveal that by analyzing the discrepancies in distribution between natural and AI-generated images, we can significantly enhance detection performance. The use of large-scale pre-trained models allows for accurate computation of predictive uncertainty, enabling us to identify images with high uncertainty as likely AI-generated. Our method achieves robust detection performance in a simple untrained manner. Overall, our approach demonstrates a promising direction for improving AI-generated image detection and mitigating potential risks associated with their misuse. Future work could delve deeper into refining the predictive models and exploring additional features that could further enhance detection accuracy.

## ETHIC STATEMENT

According to the Code of Ethics, our work does not raise any ethical concerns, as there are no human subjects, private datasets, harmful insights, or research integrity issues. Our work does not raise any ethical concerns. In particular, our work focuses on detecting AI-generated images to avoid the potential risk induced by the development of generative models.

## REPRODUCIBILITY STATEMENT

To highlight the efforts that have been made to ensure reproducibility, we summarize materials facilitating reproducible results:

- **Theoretical results.** Our work mainly focuses on the empirical investigation without explicit theatrical results.
- **Datasets.** All involved datasets are publicly available. To facilitate reproducible results, we provide details in Sec 4 and Appendix A.16.
- **Open Source.** We will release our code once the paper is accepted. If reviewers request, we will include an anonymized link to the code in our response.

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

# A APPENDIX

## A.1 NOTE ON THE REVISION

Our papers receive constructive comments from reviewers ANn9, fpAT, Ks83 and JnK7. We have revised the paper based on these comments. These comments have greatly improved our work. We would like to express our gratitude to them.

## A.2 CONCERNS ABOUT MODELS BEING CONTAMINATED BY GENERATED IMAGES

With the proliferation of generated images, future large-scale models could become contaminated by such data, making it increasingly difficult to distinguish between natural and generated images. One potential solution is to employ machine unlearning (Yao et al., 2023) techniques to detect and address generated images. Machine unlearning focuses on removing the influence of specific data from pre-trained models, primarily due to privacy concerns. In this context, when generative images are incorporated into the training process of large-scale models, we can utilize machine unlearning techniques to eliminate the effects of these images on pre-trained models. This approach would help ensure that the features of natural and generated images remain distinct and separable.

## A.3 DISCUSSION ON DISTRIBUTION DISCREPANCY

In this paper, the core assumption we make is that there is data distribution discrepancy between natural and generated images. This assumption is valid for current generative models and has been confirmed by many

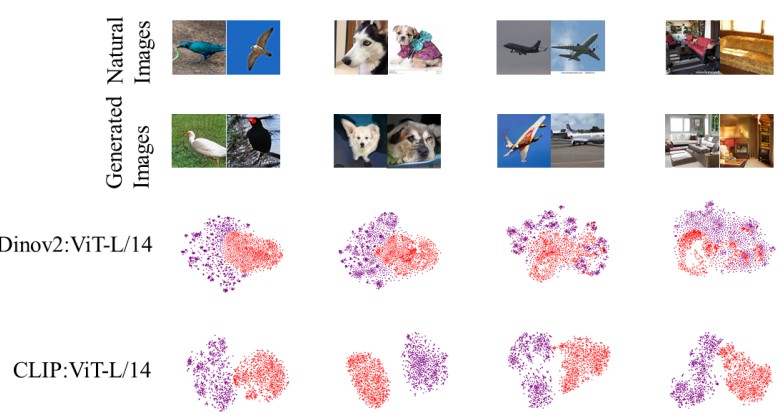

Figure 7: Feature distribution discrepancy between the generated and natural images on DINOv2 and CLIP. • and • represent the feature of natural images and AI-generated images on the corresponding models.

works (Corvi et al., 2023; Tan et al., 2024; Ricker et al., 2024). This assumption is also the foundation of many generative image detection methods (we cannot distinguish between images that are indistinguishable).

Secondly, we observe that this discrepancy in data distribution can be captured by the representation space of a vision model pre-trained on a large number of natural images, i.e., there is feature distribution discrepancy between the generated and natural images, as shown in Figure 7. However, this remains an observation, and we have not found theoretical proof despite reviewing the literature. We only observe a similar phenomenon in UnivFD (Ojha et al., 2023), where the feature distribution discrepancy is observed in the representation space of CLIP:ViT-L/14.

That said, we can confirm the existence of feature distribution discrepancy of generated and natural images based on an important metric for evaluating generative models, the FID score. The FID score measures the feature distribution discrepancy between natural and generated images on the Inception network (Szegedy et al., 2015). When the FID score is 0, it indicates that the two distributions do not differ. However, even on these simple networks such as Inception v3, advanced generative models like ADM still achieve an FID score of 11.84, not to mention that on powerful models such as DINOv2, we observe significant feature distribution discrepancy.

## A.4 MEASURING FEATURE DISTRIBUTION DISCREPANCY WITH FID SCORES

We further use the "FID" score to measure the difference in feature distribution between natural and generated images. To avoid the effects of categories, we compute the FID scores using the DINOv2 model on the LSUN-BEDROOM benchmark. For each category of images, we randomly select 5000 images for calculation. In addition to calculating the FID scores between natural images and generated images, we also calculate the FID scores between natural images and natural images. As shown in Table 6, the FID scores between natural images and generated images are significantly higher than the FID scores between natural images and natural images. Moreover, there is a clear positive correlation between the detection performance of WePe and the FID score. This result fully explains the existence of feature distribution discrepancy between natural and generated images on DINOv2, and demonstrates that WePe can effectively detect the feature distribution discrepancy.

Table 6: Measuring feature distribution discrepancy with FID scores.

| Models | Natural | ADM | StyleGAN | iDDPM | DDPM | Diffusion GAN | Unleashing Transformer | Projected GAN |
|--------|---------|-----|----------|-------|------|---------------|------------------------|---------------|
| FID score | 1.09 | 18.25 | 52.31 | 59.94 | 80.44 | 116.06 | 130.00 | 132.75 |
| AUROC | 50.00 | 73.85 | 83.50 | 86.23 | 88.84 | 94.16 | 94.18 | 95.34 |

## A.5 COMPARISON WITH GRADIENT CUFF

Gradient Cuff (Hu et al., 2024) focuses on detecting jailbreak attacks in Large Language Models (LLMs). It finds that the landscape of the refusal loss is more precipitous for malicious queries than for benign queries. And then it uses stochastic gradient estimation to estimate gradient and use the gradient norm as the decision score. Thus, we can also leverage this interesting work to identify the distribution discrepancy for detection. We use the similar way to estimate gradient as the decision score. As shown in Table 7, we surprisingly find that this method even surpasses WePe. This suggests that it is also possible to distinguish between natural and generated images by estimating the gradient.

## A.6 WEPE ON LARGE MULTI-MODAL MODELS

In addition to CLIP, we further test the performance of WePe on BLIP (Li et al., 2022). As shown in Table 8, the performance of WePe is unsatisfactory on these multimodal models, which may be due to the fact that the image features of the multimodal models are more focused on semantic information, in line with our discussions.

## A.7 PERFORMANCE ON ADVERSARIAL EXAMPLES

We further test WePe on adversarial examples. We simply add Gaussian noise (with different standard deviation $\sigma$) to the test samples to simulate the adversarial examples. We test three cases: adding noise on the natural image, adding noise on the generated image and adding noise on all images. As shown in Table 9, when noise is injected, the feature similarity between the clean model and the noisy model for the noisy image decreases, which leads to a change in the detection performance. To mitigate this effect, we can perform detection by perturbing the model multiple times and using the average similarity. The effect of noise is successfully mitigated by ensemble as shown in Table 10.

Table 7: Comparison with Gradient Cuff.

| Method | AUROC | AP |
|--------|-------|-----|
| WePe | 87.99 | 85.04 |
| WePe + Gradient Cuff | 89.36 | 90.62 |

Table 8: WePe on large multi-modal models.

| Model | AUROC | AP |
|-------|-------|-----|
| DINOv2 | 87.99 | 85.04 |
| CLIP | 77.89 | 77.90 |
| BLIP | 68.25 | 64.68 |

## A.8 CONCERNS ABOUT HARD SAMPLES

WePe relies on the model being pre-trained on a large dataset of natural images. Given the abundance of natural images, it is possible WePe may misclassify other natural images that are out-of-distribution, as AI-generated images, leading to false-negative errors. Thus, we follow previous work to leverage AUROC as the main metric the evaluate different methods. This is because AUROC is a pivotal metric for reflecting the false-positive and false-negative costs. The hard (natural) samples play a crucial role in detecting generated images. We will explore how to leverage hard samples to promote the detection performance in our future work.

Table 9: Performance on adversarial examples.

| Case | AUROC | AP |
|---|---|---|
| Clean images | 87.99 | 85.04 |
| Noisy natural images, $\sigma = 0.05$ | 81.79 | 78.99 |
| Noisy natural images, $\sigma = 0.1$ | 76.16 | 73.16 |
| Noisy natural images, $\sigma = 0.15$ | 69.53 | 66.83 |
| Noisy generated images, $\sigma = 0.05$ | 88.03 | 85.82 |
| Noisy generated images, $\sigma = 0.1$ | 88.77 | 87.14 |
| Noisy generated images, $\sigma = 0.15$ | 90.25 | 88.99 |
| Noisy natural and generated images, $\sigma = 0.05$ | 84.94 | 81.98 |
| Noisy natural and generated images, $\sigma = 0.1$ | 81.46 | 78.25 |
| Noisy natural and generated images, $\sigma = 0.15$ | 78.19 | 75.32 |

Table 10: Mitigating the effects of image noise through ensemble.

| Case | AUROC | AP |
|---|---|---|
| Clean images | 87.99 | 85.04 |
| Noisy natural images, $\sigma = 0.05, n = 1$ | 81.79 | 78.99 |
| Noisy natural images, $\sigma = 0.05, n = 5$ | 85.30 | 82.59 |
| Noisy natural images, $\sigma = 0.1, n = 1$ | 76.16 | 73.16 |
| Noisy natural images, $\sigma = 0.1, n = 5$ | 81.36 | 78.39 |
| Noisy natural images, $\sigma = 0.15, n = 1$ | 69.53 | 66.83 |
| Noisy natural images, $\sigma = 0.15, n = 5$ | 75.16 | 72.58 |

## A.9 BOOSTING PERFORMANCE ON CLIP

As mentioned above, WePe does not perform satisfactorily on vision language models such as CLIP. Since CLIP needs to unite image features and text features, the image features need to be projected into the union space, which results in the projected image features being more focused on semantic information. For this reason, we can improve the performance of WePe on CLIP by using the features before projection for detection, as shown in Table 11.

Table 11: Boosting performance on CLIP.

| Model | AUROC | AP |
|---|---|---|
| DINOv2 | 87.99 | 85.04 |
| CLIP | 77.89 | 77.90 |
| CLIP without projection matrix | 84.82 | 84.20 |

Table 12: Comparison of detection times.

| Method | Time (s) |
|---|---|
| AEROBLADE | 17.6 |
| RIGID | 3.7 |
| WePe | 4.5 |

## A.10 COMPARISON OF DETECTION TIMES.

Table 13: Evading detection of RIGID by adding noise to the generated image.

| Model | AUROC | AP |
|---|---|---|
| RIGID | 83.58 | 81.58 |
| WePe | 87.99 | 85.04 |
| RIGID with noisy generated images | *18.69* | *34.51* |
| WePe with noisy generated images | 88.77 | 87.14 |

Our method use a perturbed pretrained model that is fixed during inferring all test samples. Thus, our method can be processed within two forward passes. This is equal to the cost of RIGID that requires two forward passes for clean and noisy images. However, RIGID can concatenate clean and noisy images in a mini batch and obtain detection results by with a single forward pass. AEROBLADE requires only one forward pass, but it needs to compute the reconstruction error of the image. This takes a long time to reconstruct at the pixel level. Besides, AEROBLADE needs to use a neural network to compute the LPIPS score, leading to computational complexity. As shown in Table 12, we compare the time required to detect 100 images under the same conditions. Since AEROBLADE needs to calculate the image reconstruction error, it has the lowest detection efficiency. RIGID can obtain detection results in a single forward pass by concatenating clean and noisy images, whereas WePe requires two forward passes, which results in WePe's detection efficiency being inferior to RIGID's. However, WePe can be parallelized across two devices to obtain the detection results in a single forward pass.

### A.11 COMPARISON WITH RIGID

The main differences between WePe and RIGID are as follows:

- The approach proposed by RIGID stems only from the phenomenon it observed: namely, that natural and generated images show different sensitivities to noise in the representation space of DINOv2. Instead, WePe explicitly proposes that there is **distribution discrepancy** between natural and generated images, and utilizes the difference in uncertainty to expose distribution discrepancy between natural and generated images.

- RIGID utilizes the difference in sensitivity to noise between natural image and generated image for detection. Although the generated image is more sensitive, it is easy to think of a way to avoid detection, i.e., **the user adds noise to the generated image and then submits it for detection**. This approach easily removes the sensitivity of the generated image to noise. As shown in Table 13, after adding noise to a generated image, RIGID determines the noisy generated image as natural images. This is a fatal flaw of RIGID. However, WePe is not affected by this. WePe exposes the difference in distribution between the test image and the natural image through weight perturbation. Adding noise to the generated image further increases this difference, leading to even better detection performance.

### A.12 NOTE ON THE UNBIASED ASSUMPTION

In section 3.3, we make an assumption that the expected extracted feature by noised models is unbiased for that by the original model. We think this assumption is reasonable. Thanks to the over-parameterization of modern neural networks and advanced optimization algorithms (e.g., AdamW), it is a well-established fact that trained neural networks are usually smooth in the parameter space and show robustness to small weight perturbation (Novak et al., 2018). And this robustness is used in many applications, such as quantization (Gholami et al., 2022) and pruning (Liu et al., 2019). And in Figure 2, we also clearly show this robustness: when adding tiny noise to the model weights, the features remain almost unchanged.

### A.13 SOFTWARE AND HARDWARE

We use python 3.8.16 and Pytorch 1.12.1, and seveal NVIDIA GeForce RTX-3090 GPU and NVIDIA GeForce RTX-4090 GPU.

### A.14 WEPE WITH MULTIPLE PERTURBATION

In our experiments, taking into account the detection efficiency, we perturb the model only once, and then calculate the feature similarity of the test samples on the clean and perturbed models. We further experiment with multiple perturbations and use the mean of the feature similarity of the test samples on the clean model and all the perturbed models as the criterion for determining whether or not the image is generated by the generative models. As shown in Figure 8,

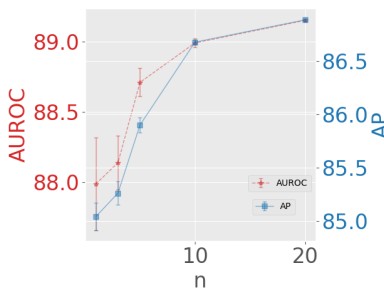

### A.15 USING NATURAL IMAGES ONLY TO SELECT WHICH LAYERS TO PERTURB

In our experiments, we use a small set of natural images and generated images to pick the parameters that need to be perturbed. When

Figure 8: WePe with multiple perturbations.

Table 14: Effect of perturbation position on natural images. We perturb each block individually, observe the similarity of features on the model of the natural image before and after the perturbation and rank these blocks.

| block | 0 | 1 | 2 | 3 | 4 | 5 | 6 | 7 | 8 | 9 | 10 | 11 | 12 | 13 | 14 | 15 | 16 | 17 | 18 | 19 | 20 | 21 | 22 | 23 |
|---|---|---|---|---|---|---|---|---|---|---|---|---|---|---|---|---|---|---|---|---|---|---|---|---|
| similarity(%) | 99.40 | 97.66 | 98.83 | 99.00 | 98.80 | 98.70 | 99.37 | 94.73 | 92.87 | 98.44 | 97.07 | 98.00 | 93.46 | 96.24 | 94.80 | 93.85 | 92.40 | 87.60 | 71.50 | 76.00 | 80.27 | 75.93 | 34.81 | 47.90 |
| rank | 1 | 9 | 4 | 3 | 5 | 6 | 2 | 13 | 16 | 7 | 10 | 8 | 15 | 11 | 12 | 14 | 17 | 18 | 22 | 20 | 19 | 21 | 24 | 23 |

Table 15: AI-generated image detection performance on ImageNet. We select the top-k blocks with the highest similarity for perturbation based on the sorting results.

| Methods | ADM | | ADMG | | LDM | | DiT | | Models BigGAN | | GigaGAN | | StyleGAN XL | | RQ-Transformer | | Mask GIT | | Average | |
|---|---|---|---|---|---|---|---|---|---|---|---|---|---|---|---|---|---|---|---|---|
| | AUROC | AP | AUROC | AP | AUROC | AP | AUROC | AP | AUROC | AP | AUROC | AP | AUROC | AP | AUROC | AP | AUROC | AP | AUROC | AP |
| *Training Methods* | | | | | | | | | | | | | | | | | | | | |
| CNNspot | 62.25 | 63.13 | 63.28 | 62.27 | 63.16 | 64.81 | 62.85 | 61.16 | 85.71 | 84.93 | 74.85 | 71.45 | 68.41 | 68.67 | 61.83 | 62.91 | 60.98 | 61.69 | 67.04 | 66.78 |
| Ojha | 83.37 | 82.95 | 79.60 | 78.15 | 80.35 | 79.71 | 82.93 | 81.72 | 93.07 | 92.77 | 87.45 | 84.88 | 85.36 | 83.15 | 85.19 | 84.22 | 90.82 | 90.71 | 85.35 | 84.25 |
| DIRE | 51.82 | 50.29 | 53.14 | 52.96 | 52.83 | 51.84 | 54.67 | 55.10 | 51.62 | 50.83 | 50.70 | 50.27 | 50.95 | 51.36 | 55.95 | 54.83 | 52.58 | 52.10 | 52.70 | 52.18 |
| NPR | 85.68 | 80.86 | 84.34 | 79.79 | 91.98 | 86.96 | 86.15 | 81.26 | 89.73 | 84.46 | 82.21 | 78.20 | 84.13 | 78.73 | 80.21 | 73.21 | 89.61 | 84.15 | 86.00 | 80.84 |
| *Training-free Methods* | | | | | | | | | | | | | | | | | | | | |
| AEROBLADA | 55.61 | 54.26 | 61.57 | 56.58 | 62.67 | 60.93 | 85.88 | 87.71 | 44.36 | 45.66 | 47.39 | 48.14 | 47.28 | 48.54 | 67.05 | 67.69 | 48.05 | 48.75 | 57.87 | 57.85 |
| RIGID | 87.16 | 85.08 | 80.09 | 77.07 | 72.43 | 69.30 | 70.40 | 65.94 | 90.08 | 89.26 | 86.39 | 84.11 | 86.32 | 85.44 | 90.06 | 88.74 | 89.30 | 89.25 | 83.58 | 81.58 |
| WePe top-8 | 89.25 | 86.53 | 82.66 | 78.08 | 79.29 | 73.88 | 78.53 | 72.48 | 93.90 | 92.61 | 92.07 | 89.65 | 93.06 | 91.26 | 92.68 | 89.84 | 89.85 | 86.91 | 87.92 | 84.59 |
| WePe top-10 | 89.57 | 86.67 | 82.62 | 79.33 | 78.95 | 74.42 | 77.15 | 72.29 | 92.65 | 91.36 | 91.91 | 90.60 | 93.77 | 92.71 | 93.17 | 91.76 | 88.42 | 86.46 | 87.58 | 85.07 |
| WePe top-12 | 89.23 | 87.86 | 84.38 | 81.19 | 78.63 | 74.13 | 75.33 | 70.50 | 94.29 | 93.81 | 92.53 | 91.71 | 94.64 | 94.32 | 93.15 | 92.15 | 89.90 | 88.22 | 88.01 | 85.99 |
| WePe top-14 | 89.69 | 87.57 | 82.60 | 79.24 | 79.69 | 76.06 | 76.74 | 71.26 | 93.05 | 92.30 | 92.45 | 91.23 | 94.71 | 94.78 | 94.96 | 94.22 | 89.44 | 88.14 | 88.15 | 86.09 |
| WePe top-16 | 90.58 | 89.40 | 84.80 | 82.08 | 80.28 | 76.54 | 76.57 | 72.88 | 92.81 | 92.55 | 92.11 | 91.10 | 92.89 | 92.72 | 93.05 | 92.26 | 91.46 | 90.60 | 88.28 | 86.68 |
| WePe top-18 | 90.02 | 87.83 | 83.39 | 80.58 | 79.12 | 74.64 | 76.18 | 71.12 | 91.82 | 91.36 | 92.26 | 91.71 | 93.77 | 93.39 | 93.68 | 92.89 | 89.12 | 87.57 | 87.71 | 85.68 |

all the generated images are not available, we can also use only the real images to select the layers that need to be perturbed. Specifically, we first perturb each block alone and calculate the similarity of the features on the model of the natural image before and after the perturbation, as shown in Table 14. We then sort the similarity and select the blocks with the highest similarity for perturbation. As shown in Table 15, selecting the parameters to be perturbed in this way also achieves good performance and has strong robustness.

## A.16 DETAILS OF DATASETS

**IMAGENET.** The real images and generated images can be obtained at `https://github.com/layer6ai-labs/dgm-eval`. The images are provided by (Stein et al., 2023). The generative model includes: ADM, ADMG, BigGAN, DiT-XL-2, GigaGAN, LDM, StyleGAN-XL, RQ-Transformer and Mask-GIT. The resolution of real images and generated images are $256 \times 256$. We crop the image randomly to $224 \times 224$ resolution.

**LSUN-BEDROOM.** The real images and generated images can be obtained at `https://github.com/layer6ai-labs/dgm-eval`. The images are provided by (Stein et al., 2023). The generative model includes: ADM, DDPM, iDDPM, StyleGAN, Diffusion-Projected GAN, Projected GAN and Unleashing Transformers. The resolution of real images and generated images are $256 \times 256$. We crop the image randomly to $224 \times 224$ resolution.

**GenImag** The real images and generated images can be obtained at `https://github.com/GenImage-Dataset/GenImage`. The images are provided by (Zhu et al., 2023). The generative model includes: Midjourney, SD V1.4, SD V1.5, ADM, GLIDE, Wukong, VQDM and BigGAN. The real images come from ImageNet, and different images have different resolutions. Following (Stein et al., 2023), we resize the image to $256 \times 256$ resolution and adjust its format to keep the same with the generated images, then we randomly crop it to $224 \times 224$ resolution to extract features. For the generated images, we report our processing in detail as follows:

- Midjourney. The resolution of images generated by Midjourney is $1024 \times 1024$, and we randomly crop them to $224 \times 224$ resolution.
- SD V1.4. The resolution of images generated by SD V1.4 is $512 \times 512$, and we randomly crop them to $224 \times 224$ resolution.
- SD V1.5. The resolution of images generated by SD V1.5 is $512 \times 512$, and we randomly crop them to $224 \times 224$ resolution.
- ADM. The resolution of images generated by SD V1.5 is $256 \times 256$, and we randomly crop them to $224 \times 224$ resolution.
- GLIDE. The resolution of images generated by SD V1.5 is $256 \times 256$, and we randomly crop them to $224 \times 224$ resolution.
- Wukong. The resolution of images generated by SD V1.5 is $512 \times 512$, and we randomly crop them to $224 \times 224$ resolution.
- VQDM. The resolution of images generated by SD V1.5 is $256 \times 256$, and we randomly crop them to $224 \times 224$ resolution.
- BigGAN. The resolution of images generated by SD V1.5 is $128 \times 128$, and we fill them with zero pixels to $224 \times 224$ resolution.

