# OpenReview forum: "Detecting Discrepancies Between Generated and Natural Images Using Uncertainty"
_ICLR.cc/2025/Conference — Submitted to ICLR 2025_

### Official Review · Reviewer_JnK7 · 2024-11-02

**Soundness:** 1
**Presentation:** 1
**Contribution:** 2
**Rating:** 3
**Confidence:** 5

**Summary:**

This paper proposes a novel method for detecting AI-generated content by leveraging the distributional discrepancy between real and generated images. Unlike most existing works, this detection method is training-free. Specifically, it uses a pretrained SSL vision foundation model, DINOv2, to estimate predictive uncertainty with respect to model weight perturbation and filters AI-generated images through thresholding. Empirical results show that this method achieves state-of-the-art or comparable performance across datasets.

**Strengths:**

1. The use of model perturbation and Bayesian inference is an interesting approach.

2. The proposed method is efficient, as it requires no training or additional data to facilitate AI-generated content detection.

3. The experimental results demonstrate the potential of the method. It consistently performs competitively across different datasets, and ablation studies show that it is robust across various configurations.

4. The paper is generally well-written, with only minor typos and typesetting errors.

**Weaknesses:**

1. The proposed method is less efficient than its training-free counterparts. For example, RIGID applies multiple input perturbations to the original image to obtain detection scores, while AEROBLADE uses reconstruction errors for thresholding; both methods can be processed in minibatches and within a single forward pass.

2. The method lacks novelty and proper theoretical analysis and justification. It essentially uses the same criterion as RIGID for "predictive uncertainty" (i.e., cosine similarity), except that it perturbs weights instead of inputs. The approximation, $ 2 - \frac{2}{n}\sum_{k=1}^n f(x; \theta_k)^T f(x; \theta) $, can be interpreted as two times the **average cosine distance** (i.e., one minus the average cosine similarity) between features extracted by the noised models and those by the original model. While the predictive uncertainty concept seems novel initially, it ultimately just switches from input perturbation to weight perturbation.

3. The paper overclaims the generalizability of the proposed method compared to its training-free counterparts, while it still relies on the strong assumption that the feature distributions of real and generated images differ significantly.

4. The derivation lacks detail and does not hold in general cases. In Equation (3), the final equality holds only if $ ||f(x, \theta_{*}) ||_2 = 1$, which is true only when using **L2-normalized** features extracted by DINOv2. This technical detail should be highlighted in the main text. Additionally, the assumption that the expected extracted feature by noised models is unbiased for that by the original model is unlikely to hold, as neural networks are highly nonlinear, making linear expectations improbable.

5. The empirical performance is unconvincing. Although more complex than RIGID, consuming more memory (requiring at least two model copies, if not more), requiring additional hyperparameter tuning, and being slower than other training-free baselines (as the functional evaluations of this method cannot be efficiently parallelized), it still fails to outperform the baselines with meaningful margins in many cases.

**Questions:**

See weaknesses.

---

> ### Author Response · Authors · 2024-11-22
>
> We sincerely appreciate your dedicated time reviewing our paper and are grateful for your constructive comments. According to your constructive comments, we provide detailed feedback below, which has been added into our revision. We hope our feedback can address your concerns and improve our work.
>
> >Q1: The proposed method is less efficient than its training-free counterparts. For example, RIGID applies multiple input perturbations to the original image to obtain detection scores, while AEROBLADE uses reconstruction errors for thresholding; both methods can be processed in minibatches and within a single forward pass.
>
> A1: We apologize for the misunderstanding. The efficiency of our method is equal to (RIGID) or better than that of training-free counterparts (AEROBLADE).
> - Our method use a perturbed pre-trained model that is fixed during inferring all test samples. Thus, our method can be processed in minibatches and within two forward passes. This is equal to the cost of RIGID that requires two forward passes for clean and noisy images. AEROBLADE requires only one forward pass, but it needs to compute the reconstruction error of the image. This takes a long time to reconstruct at the pixel level. Besides, AEROBLADE needs to use a neural network to compute the LPIPS score, leading to computational complexity.
>
> - As shown in the table below, we compare the time of the three methods for detecting 100 images under the same conditions. Aligning with the above discussions, since AEROBLADE needs to calculate the image reconstruction error, and RIGID needs to add noise to the image before extracting the features during the test, whereas WePe extract the features directly, WePe achieves the best detection efficiency.
>
> |           | time  |
> |-----------|-------|
> | AEROBLADE | 17.6s |
> | RIGID     | 7.2s  |
> | WePe      | 4.5s  |
>
> Thanks again for your suggestion and we will incorporate the comparison results in the revised version.
>
> >Q2: The method lacks novelty and proper theoretical analysis and justification. It essentially uses the same criterion as RIGID for "predictive uncertainty" (i.e., cosine similarity), except that it perturbs weights instead of inputs. The approximation,  can be interpreted as two times the average cosine distance (i.e., one minus the average cosine similarity) between features extracted by the noised models and those by the original model. While the predictive uncertainty concept seems novel initially, it ultimately just switches from input perturbation to weight perturbation.
>
> A2 : We would like to clarify what makes us different from Rigid as follows:
> *  The approach proposed by RIGID stems only from the phenomenon it observed: namely, that natural and generated images show different sensitivities to noise in the representation space of DINOv2. Instead, WePe explicitly proposes that there is **distribution discrepancy** between natural and generated images, and utilizes the difference in uncertainty to expose distribution discrepancy between natural and generated images.
> * RIGID utilizes the difference in sensitivity to noise between  natural image and generated image for detection. Although the generated image is more sensitive, it is easy to think of a way to avoid detection, i.e., **the user adds noise to the generated image and then submits it for detection**. This approach easily removes the sensitivity of the generated image to noise. As shown in the table below, after adding noise to a generated image, RIGID determines the noisy generated image as natural images. This is a fatal flaw of RIGID. However, WePe is not affected by this. WePe exposes the difference in distribution between the test image and the natural image through weight perturbation. Adding noise to the generated image further increases this difference, leading to even better detection performance.
>
> |                                   | AUROC | AP    |
> |-----------------------------------|-------|-------|
> | RIGID                             | 83.58 | 81.58 |
> | WePe                              | 87.99 | 85.04 |
> | RIGID with noisy generated images | *18.69* | *34.51* |
> | WePe with noisy generated images  | 88.77 | 87.14 |

---

> ### Author Response · Authors · 2024-11-22
>
> >Q3: The paper overclaims the generalizability of the proposed method compared to its training-free counterparts, while it still relies on the strong assumption that the feature distributions of real and generated images differ significantly.
>
> A3: We apologize for the misunderstanding.
> First, the generalizability of the proposed method is demonstrated by our experimental results in Tables 1, 2, and 3.
> Second, it seems to be impossible to distinguish samples from two distributions without discrepancy. Thus, we introduce the assumption that **the feasibility of distinguishing natural images from AI-generated ones is grounded in the distribution discrepancy between them**. Note that, our assumption underscores the discrepancy between data distribution rather than the mentioned feature distributions.
> Third, our method is indeed based on the discrepancy between natural and generated image distributions. Thus, we propose to use predictive uncertainty to identify the discrepancy for detecting generated images.
>
> >Q4: The derivation lacks detail. In Eq. (3), the final equality holds when using L2-normalized features extracted by DINOv2. This technical detail should be highlighted in the main text.
>
> A4: Following your valuable suggestion, we will highlight the detail in the main text.
>
> >Q5: The derivation does not hold in general cases. The assumption that the expected extracted feature by noised models is unbiased for that by the original model is unlikely to hold.
>
> A5: Thanks for your comments. We believe the proposed assumption holds. Thanks to the over-parameterization of modern neural networks and advanced optimization algorithms (e.g., AdamW), it is a well-established fact that trained neural networks are usually smooth in the parameter space and show robustness to small weight perturbation [r1]. And this robustness is used in many applications, such as quantization [r2] and pruning [r3]. And our Figure2 also clearly shows this robustness: when adding tiny noise to the model weights, the features remain almost unchanged.
>
> >Q6: The empirical performance is unconvincing. Although more complex than RIGID, consuming more memory (requiring at least two model copies, if not more), requiring additional hyperparameter tuning, and being slower than other training-free baselines (as the functional evaluations of this method cannot be efficiently parallelized), it still fails to outperform the baselines with meaningful margins in many cases.
>
> A6: We apologize for the misunderstanding. First, we would like to highlight that our method is not more complex than the frequently mentioned RIGID method. Second, our method can outperform the frequently mentioned RIGID in all benchmarks. Detailed explanations are as follows.
>
> First, our method achieves the same memory cost, i.e., two models v.s. two samples. Moreover, the parameter DINOv2-L/14 used in our experiments is only 300M, which is obviously easy to load even if WePe needs to load two models. We merely introduce two hyper-parameters: the degree of perturbation and the index of perturbed blocks. All introduced hyper-parameters are the same in all experiments. For computational efficiency, we have clarified earlier that our method is efficient and can be easily parallelized .
> Second, our method outperforms RIGID by 4.41%, 2.81%, and 3.08% on ImageNet, LSUN-BEDROOM and GenImage, respectively. Moreover, the performance gain is more significant when images are degraded using different perturbations.
>
> Reference:
>
> [r1]: Sensitivity and generalization in neural networks: an empirical study
>
> [r2]: A survey of quantization methods for efficient neural network inference
>
> [r3]: Rethinking the value of network pruning

---

> > ### Comment · Reviewer_JnK7 · 2024-11-25
> >
> > Thank you for your detailed responses and for providing new findings. I appreciate your effort in addressing my concerns and offering additional experimental results. Below are a few points that I would like further clarification on:
> >
> > > This is equal to the cost of RIGID that requires two forward passes for clean and noisy images.
> >
> > **Q1:** Could the authors clarify why two forward passes are necessary for RIGID? Since clean and noisy images can be concatenated and processed in a single mini-batch during one forward pass, wouldn’t this reduce computational overhead?
> >
> > > RIGID needs to add noise to the image before extracting the features during the test, whereas WePe extract the features directly, WePe achieves the best detection efficiency.
> >
> > **Q2:** If I understand correctly, WePe requires adding noise to the model weights before extracting features. How, then, does it "extract the features directly" while achieving "the best detection efficiency"?
> >
> > > the user adds noise to the generated image and then submits it for detection
> >
> > **Q3:** This adversarial setting is quite intriguing! Could the authors elaborate on the specific configuration used for this experiment? For instance, what noise scale was applied to achieve the results presented in Table 12?
> > Additionally:
> >
> > - How would RIGID perform if noise were added to both natural and generated images, as in the experiments that improved detection results for WePe (e.g., Table 8)? Would this approach also enhance RIGID's detection performance?
> > - Would a similar phenomenon occur if the DINOv2 model itself were perturbed by an adversary? For example, if noise were added to the model parameters, would the detection rates for natural and generated images remain stable or degrade?
> >
> > > Thanks to the over-parameterization of modern neural networks and advanced optimization algorithms (e.g., AdamW), it is a well-established fact that trained neural networks are usually smooth in the parameter space and show robustness to small weight perturbation [r1].
> >
> > **Q4:** While modern neural networks are indeed smooth and robust to small weight perturbations, the authors acknowledged that "when the noise level is small, the features obtained before and after the model perturbation are extremely similar, ... will result in the inability to effectively differentiate between real and generated images." In practice, wouldn't a noise level large enough for effective detection violate this smoothness assumption? Could you provide empirical evidence to demonstrate that the assumption holds under the noise levels used in your experiments?

---

> > > ### Author Response · Authors · 2024-11-25
> > >
> > > Thanks for your prompt reply!
> > >
> > > >Q1: Could the authors clarify why two forward passes are necessary for RIGID? Since clean and noisy images can be concatenated and processed in a single mini-batch during one forward pass, wouldn’t this reduce computational overhead?
> > >
> > > A1: For each test image, RIGID requires both the original and noisy images to derive the similarity. The computational cost is equal to a two forward passes for the original images. We agree that RIGID can detect in a single forward pass scheme by concatenating original and noisy images. In this regard, our method can be parallelized across two devices to obtain the detection results in a single forward pass.
> > >
> > > The mismatch between our responses and your understanding may result from our configuration, i.e., a large batch size. Following your suggestion, we re-run the experiment as suggested, and the results are shown in the table below. RIGID is indeed more efficient than WePe in this scenario.
> > >
> > > |       | time |
> > > |-------|------|
> > > | RIGID | 3.7s |
> > > | WePe  | 4.5s |
> > >
> > > >Q2: If I understand correctly, WePe requires adding noise to the model weights before extracting features. How, then, does it "extract the features directly" while achieving "the best detection efficiency"?
> > >
> > > A2: We apologize for the misunderstanding. We perturb the model in advance before conducting the detection. During the detection process, we don't modify the model but simply extract features using the pre-trained and perturbed models. We will clarify this point in the revision to avoid misunderstandings.
> > >
> > > >Q3: This adversarial setting is quite intriguing! Could the authors elaborate on the specific configuration used for this experiment? For instance, what noise scale was applied to achieve the results presented in Table 12?
> > >
> > > A3: Thanks for your advice. We simply add Gaussian noise with a noise intensity of 0.1 to the generated image. We will detail this in the revision.
> > >
> > > >Q4: How would RIGID perform if noise were added to both natural and generated images, as in the experiments that improved detection results for WePe (e.g., Table 8)? Would this approach also enhance RIGID's detection performance?
> > >
> > > A4: Thank you for your comments. Following your suggestion, we further conduct experiments on adding noise to both natural and generated images. The results are shown in the table below, and this method improves the detection performance of RIGID. However, when the noise intensity added to the generated image exceeds that added to the natural image, the performance of RIGID remains suboptimal.
> > >
> > > |                                                                     | RIGID |       | WePe  |       |
> > > |---------------------------------------------------------------------|-------|-------|-------|-------|
> > > | case                                                                | AUROC | AP    | AUROC | AP    |
> > > | Clean images                                                        | 83.58 | 81.58 | 87.99 | 85.04 |
> > > | Noisy generated images with $\sigma = 0.1$                              | 18.69 | 34.51 | 88.77 | 87.14 |
> > > | Noisy natural and generated images with $\sigma = 0.1$                  | 79.49 | 75.49 | 81.46 | 78.25 |
> > > | Noisy natural images with $\sigma = 0.1$ and Noisy generated images with $\sigma = 0.2$ | 67.68 | 64.07 | 84.26 | 82.96 |
> > >
> > > >Q5: Would a similar phenomenon occur if the DINOv2 model itself were perturbed by an adversary? For example, if noise were added to the model parameters, would the detection rates for natural and generated images remain stable or degrade?
> > >
> > > A5: Thank you for your valuable insights. However, we believe that such a situation is unlikely because malicious users typically cannot access or modify the parameters of the detection model.
> > >
> > > In response to your time and efforts, we conduct further experiments where the model is perturbed as you described. The perturbation strength is set to 0.1, and the experimental results are as follows. It demonstrates that WePe's performance remains stable under such perturbation.
> > >
> > > |                 | AUROC | AP    |
> > > |-----------------|-------|-------|
> > > | clean model     | 87.99 | 85.04 |
> > > | perturbed model | 86.14 | 83.87 |

---

> > > ### Author Response · Authors · 2024-11-25
> > >
> > > >Q6: While modern neural networks are indeed smooth and robust to small weight perturbations, the authors acknowledged that "when the noise level is small, the features obtained before and after the model perturbation are extremely similar, ... will result in the inability to effectively differentiate between real and generated images." In practice, wouldn't a noise level large enough for effective detection violate this smoothness assumption? Could you provide empirical evidence to demonstrate that the assumption holds under the noise levels used in your experiments?
> > >
> > > A6: We apologize for the misunderstanding.
> > > By “when the noise level is small”, we mean a noise intensity of less than 0.01, at which point the test images show minimal feature changes on the model before and after the perturbation. In our experiments, we use a noise intensity of 0.1. As shown in Figure2 (middle column), the features of both natural and generated images also show little change when the model is perturbed with a noise intensity of 0.1.
> > >
> > > To further address your concerns, we quantify the cosine similarity of the features on the model before and after perturbation for the test samples at different noise levels. The results are presented in the following table. When the noise intensity is 0.01, the cosine similarity reaches 1, indicating that the features are almost unchanged. When the noise intensity is increased to 0.03, the feature similarity for natural images is 0.999, while for generated images it is 0.995. Although these changes are still minimal, WePe has achieved good detection performance at this noise level.
> > >
> > > We will include the above discussion in the revision to further support our argument.
> > >
> > > | noise intensity                        | 0.01  | 0.03  | 0.05  | 0.1   | 0.15  | 0.2    |
> > > |----------------------------------------|-------|-------|-------|-------|-------|--------|
> > > | Feature similarity of natural images   | 1     | 0.999 | 0.997 | 0.985 | 0.946 | 0.8984 |
> > > | Feature similarity of generated images | 1     | 0.995 | 0.987 | 0.948 | 0.855 | 0.729  |
> > > | AUROC                                  | 68.85 | 86.98 | 87.47 | 87.99 | 86.21 | 83.57  |
> > > | AP                                     | 68.80 | 82.78 | 85.10 | 85.04 | 83.89 | 81.29  |
> > >
> > > Frequently mentioning RIGID is informative, making us deeply understand the reviewer's responsibility. Thus, we sincerely appreciate your contribution to reviewing our work and improving its quality.  We hope our responses can address your concerns and look forward to any detailed and constructive comments to enhance the quality of our work further.

---

> ### Author Response · Authors · 2024-11-26
>
> Dear Reviewer #JnK7,
>
> Thank you very much for your time and valuable feedback.
>
> Here is a summary of our response for your convenience:
>
> **(1) Issues with computational overhead:** We've clarified and re-compared it.
>
> **(2) Issues with WePe's detection process:** WePe perturbs the model in advance before detection and does not modify the model during the detection process.
>
> **(3) Issues with configuration:** We add Gaussian noise of intensity 0.1 to the generated images.
>
> **(4) Issues with adding noise to both natural and generated images simultaneously:** Adding noise to both natural and generated images can mitigate the degradation of RIGID's performance. However, RIGID still performs poorly when the noise intensity of the generated image is higher than that of the natural image.
>
> **(5) Issues with the model being perturbed:** We have conducted experiments where the model is perturbed and have shown the robustness.
>
> **(6) Issues with the assumption of unbiasedness:** We have provided further justification for the validity of this assumption, which is supported by experimental results.
>
> We understand you're busy. But as the window for responding and paper revision is closing, would you mind checking our response and confirm whether you have any further questions? We are very glad to provide answers and revision to your further questions.
>
> Best regards and thanks,
>
> Authors of #13682

---

> ### Author Response · Authors · 2024-11-27
>
> Dear Reviewer #JnK7,
>
> Thanks a lot for your time in reviewing and insightful comments, according to which we have carefully revised the paper to answer the questions. We sincerely understand you’re busy. However, as the paper revision window is about to close, would you mind checking the response and revision to confirm where you have any further questions?
>
> We are looking forward to your reply and happy to answer your further questions.
>
> Best regards
>
> Authors of #13682

---

> ### Author Response · Authors · 2024-11-29
>
> Dear Reviewer #JnK7,
>
> Thanks a lot for your time in reviewing and insightful comments. We sincerely understand you're busy. But as the window for discussion is closing, could you please take some time to review our responses? We'd love to get your further feedback and do our best to address it to enhance our work.
>
> We are looking forward to your reply!
>
> Best regards,
>
> Authors of #13682

---

> ### Author Response · Authors · 2024-11-30
> **Welcome for more discussions**
>
> Dear Reviewer #JnK7,
>
> Thanks a lot for your time in reviewing and insightful comments. We sincerely understand you're busy. But as the window for discussion is closing, could you please take some time to review our responses? We'd love to get your further feedback and do our best to address it to enhance our work.
>
> We are looking forward to your reply!
>
> Best regards,
>
> Authors of #13682

---

> ### Author Response · Authors · 2024-12-01
>
> Dear Reviewer #JnK7,
>
> Thanks a lot for your time in reviewing and insightful comments. We sincerely understand you're busy. But as the window for discussion is closing, could you please take some time to review our responses? We'd love to get your further feedback and do our best to address it to enhance our work.
>
> We are looking forward to your reply!
>
> Best regards,
>
> Authors of #13682

---

> ### Author Response · Authors · 2024-12-03
> **Window for discussion is closing**
>
> Dear Reviewer JnK7,
>
> Thanks a lot for your time in reviewing and insightful comments. We sincerely understand you're busy. But as the window for discussion is closing, could you please check to see if your concerns have been addressed? We'd love to get your further feedback and do our best to address it to enhance our work.

---

> ### Author Response · Authors · 2024-12-04
>
> Dear Reviewer JnK7,
>
> Could you please check to see if there are any problems that haven't been resolved?

---

### Official Review · Reviewer_Ks83 · 2024-11-04

**Soundness:** 3
**Presentation:** 3
**Contribution:** 3
**Rating:** 6
**Confidence:** 3

**Summary:**

This paper aims to detect AI-generated images from natural images by leveraging predictive uncertainty, which offering an effective approach for capturing distribution shifts. In order to ensure that the model has been trained over sufficient natural images, this paper leverages large-scale pre-trained models to calculate the uncertainty.

**Strengths:**

1. This work presents an intriguing approach by leveraging the predictive uncertainty of the model to detect AI-generated images.
2. The paper conducts thorough experiments to test the validity of the proposed method.
3. This paper is well-written and easy to follow.

**Weaknesses:**

1. The abstract does not mention the sensitivity of the samples to the weight perturbation of the large model.
2. The method proposed in this paper relies on the model being pre-trained on a large dataset of natural images. Given the abundance of natural images, it raises the question of whether this method might misclassify other natural images that are out-of-distribution, as AI-generated images.
3. The paper only selects the DINOv2 as the large-scale pre-trained model. Although it discusses the reasons for choosing DINOv2 and not using CLIP, it is difficult to convince that the proposed method, WePe, is applicable to other large-scale pre-trained models.

**Questions:**

Please see the weaknesses.

---

> ### Author Response · Authors · 2024-11-22
>
> We sincerely appreciate your dedicated time reviewing our paper and are grateful for your constructive comments. According to your constructive comments, we provide detailed feedback below, which has been added into our revision. We hope our feedback can address your concerns and improve our work.
>
> >Q1: The abstract does not mention the sensitivity of the samples to the weight perturbation of the large model.
>
> A1: Thanks for your valuable suggestion. We have revised the abstract by adding the mentioned information as follows.
>
> *Inspired by MC-Dropout, we perturb pre-trained models and find that the uncertainty can be captured by perturbing the weights of pre-trained models.*
>
> >Q2: The method proposed in this paper relies on the model being pre-trained on a large dataset of natural images. Given the abundance of natural images, it raises the question of whether this method might misclassify other natural images that are out-of-distribution, as AI-generated images.
>
> A2: Thanks for your valuable comments. The situation you are concerned about does exist. Pre-trained model would misclassify some natural images as generated images, leading to false-negative errors. Thus, we follow previous work to leverage AUROC as the main metric the evaluate different methods. This is because AUROC is a pivotal metric for reflecting the false-positive and false-negative costs. Inspired by your comments, we believe that hard (natural) samples play a crucial role in detecting generated images. Thus, we will explore how to leverage hard samples to promote the detection performance in our future work.
>
> >Q3: The paper only selects the DINOv2 as the large-scale pre-trained model. Although it discusses the reasons for choosing DINOv2 and not using CLIP, it is difficult to convince that the proposed method, WePe, is applicable to other large-scale pre-trained models.
>
> A3: Thanks for your constructive comments. Following your valuable comments, we evaluate the detection performance of WePe using CLIP (ViT-L/14), and list the results in the following table.  It is shown that the performance of WePe on the vision model is better than on the vision language model, in line with the discussion in our paper. Since CLIP needs to unite image features and text features, the image features need to be projected into the union space, which results in the projected image features being more focused on semantic information. For this reason, we can improve the performance of WePe on CLIP by using the features before projection for detection, as shown in the table below. These results illustrate that WePe is effective on various models.
>
> |        | AUROC | AP    |
> |--------|-------|-------|
> | CLIP   | 77.89 | 77.90 |
> | DINOv2 | 87.99 | 85.04 |
> | CLIP (without projection matrix)   | 84.82 | 84.20 |

---

> ### Author Response · Authors · 2024-11-25
>
> Dear Reviewer #Ks83,
>
> Thank you very much for your time and valuable feedback.
>
> Here is a summary of our response for your convenience:
>
> **(1) Issues of missing information:** We have added the relevant information to the abstract.
>
> **(2) Issues of hard samples:** Given the abundance of natural images, WePe may misclassify some hard natural images as generated images, leading to false-negative errors. This issue is reflected in the AUROC metric. We will explore how to leverage hard samples to improve detection performance in our future work.
>
> **(3) Issues in the performance of WePe on other large-scale pre-trained models:** We have further validated the effectiveness of WePe on CLIP.
>
> We understand you're busy. But as the window for responding and paper revision is closing, would you mind checking our response and confirm whether you have any further questions? We are very glad to provide answers and revision to your further questions.
>
> Best regards and thanks,
>
> Authors of #13682

---

> ### Author Response · Authors · 2024-11-27
>
> Dear Reviewer #Ks83,
>
> Thanks a lot for your time in reviewing and insightful comments, according to which we have carefully revised the paper to answer the questions. We sincerely understand you’re busy. However, as the paper revision window is about to close, would you mind checking the response and revision to confirm where you have any further questions?
>
> We are looking forward to your reply and happy to answer your further questions.
>
> Best regards
>
> Authors of #13682

---

> ### Author Response · Authors · 2024-11-29
>
> Dear Reviewer #Ks83,
>
> Thanks a lot for your time in reviewing and insightful comments. We sincerely understand you're busy. But as the window for discussion is closing, could you please take some time to review our responses? We'd love to get your further feedback and do our best to address it to enhance our work.
>
> We are looking forward to your reply!
>
> Best regards,
>
> Authors of #13682

---

> ### Author Response · Authors · 2024-11-30
> **Welcome for more discussions**
>
> Dear Reviewer #Ks83,
>
> Thanks a lot for your time in reviewing and insightful comments. We sincerely understand you're busy. But as the window for discussion is closing, could you please take some time to review our responses? We'd love to get your further feedback and do our best to address it to enhance our work.
>
> We are looking forward to your reply!
>
> Best regards,
>
> Authors of #13682

---

> ### Author Response · Authors · 2024-12-01
>
> Dear Reviewer #Ks83,
>
> Thanks a lot for your time in reviewing and insightful comments. We sincerely understand you're busy. But as the window for discussion is closing, could you please take some time to review our responses? We'd love to get your further feedback and do our best to address it to enhance our work.
>
> We are looking forward to your reply!
>
> Best regards,
>
> Authors of #13682

---

> > ### Comment · Reviewer_Ks83 · 2024-12-03
> >
> > Thank you so much for diligently responding to my opinion. I have read your review, and I will increase my rating to 6. Thank you.

---

> > > ### Author Response · Authors · 2024-12-03
> > >
> > > Thanks for your reply despite such a busy period. We sincerely appreciate that you can raise the score. If you have any more questions, please feel free to bring it up and we'll do our best to address it!

---

### Official Review · Reviewer_fpAT · 2024-11-04

**Soundness:** 3
**Presentation:** 4
**Contribution:** 3
**Rating:** 5
**Confidence:** 4

**Summary:**

The paper introduces a novel approach for detecting AI-generated images by leveraging predictive uncertainty to mitigate misuse and associated risks. The proposed method is grounded in the distribution discrepancy between natural and AI-generated images, utilizing predictive uncertainty to capture shifts in distributions. The authors advocate using large-scale pre-trained models to compute the uncertainty scores, identifying images that induce higher uncertainty as potentially AI-generated. The paper's contributions are demonstrated through comprehensive experiments across multiple benchmarks, showing the effectiveness of the proposed method.

**Strengths:**

1. The technical claims presented are sound, supported by comprehensive experiments and a thorough analysis of predictive uncertainty.
2. The paper is well-written, and the authors make a clear case for their method.
3. Given the increasing concern over deepfakes and manipulated content, the research is timely and addresses a significant societal issue.

**Weaknesses:**

1. The authors did not evaluate the proposed models on large multi-modal models (LMMs) such as CLIP. Given the growing popularity of using text information to assist in image generation, there is a practical and pressing need for effective methods to detect images generated by these LMMs. Please refer to question 2 for further details.
2. The author should study the predictive effectiveness of uncertainty. If the model's predictive performance improves as the number of samples n increases, this would indirectly prove the unbiased nature of the method proposed in the article in terms of predictive uncertainty. See question 3.
3. The authot doesn’t test the method’s performance on adversarial examples. See question 4.

**Questions:**

Question 1: The uncertainty estimation method, based on weight perturbation and multiple queries, reminds me of gradient estimation, could the author compare the proposed method with Gradient Cuff [1]? Though this method is designed for text originally, you can borrow the idea from it, just using the﻿ as the function value and estimate the gradient based on the perturb noise added to the weight.
Question 2: Could the author show the method’s performance on CLIP or other multi-modality image-generation models? I really understand the results may be sub-optimal, so not performing well on those models won’t affect my rating. Just curious how this method performs on LMMs.
Question 3: Could the author show the method’s scaling performance with increased n?
Question 4: Since the key technical contribution of this method is weight perturbation, I’m interested to see how this method performs on adversarial examples? The author can get the adversarial examples by simply applying noises to the test images.
By addressing these points, the authors can further solidify their contribution and provide a more comprehensive understanding of their work. I would adjust my rating accordingly.
references
[1] Gradient Cuff: Detecting Jailbreak Attacks on Large Language Models by Exploring Refusal Loss Landscapes. Xiaomeng Hu, Pin-Yu Chen, Tsung-Yi Ho

---

> ### Author Response · Authors · 2024-11-22
>
> We sincerely appreciate your dedicated time reviewing our paper and are grateful for your constructive comments. According to your constructive comments, we provide detailed feedback below, which has been added into our revision. We hope our feedback can address your concerns and improve our work.
>
> >Q1: The uncertainty estimation method, based on weight perturbation and multiple queries, reminds me of gradient estimation, could the author compare the proposed method with Gradient Cuff [r1]? Though this method is designed for text originally, you can borrow the idea from it, just using the as the function value and estimate the gradient based on the perturb noise added to the weight.
>
> A1: Thanks for your constructive suggestion and mentioning this interesting work! Gradient Cuff finds that the landscape of the refusal loss is more precipitous for malicious queries than for benign queries. And then it uses stochastic gradient estimation to estimate gradient and use the gradient norm as the decision score. Thus, we can also leverage the interesting work to identify the distribution discrepancy for detection.
> Following your suggestion, we use the similar way to estimate gradient as the decision score. As shown in the following table, we surprisingly find that this method even surpasses WePe. This suggests that it is also possible to distinguish between natural and generated images by estimating the gradient. Your insights provide a possible link between these two promising directions, and we'll incorporate the above discussion into the revision.
>
> |               | AUROC | AP    |
> |---------------|-------|-------|
> | WePe + Gradient Cuff | 89.36 | 90.62 |
> | WePe          | 87.99 | 85.04 |
>
> >Q2: Could the author show the method’s performance on CLIP or other multi-modality image-generation models? I really understand the results may be sub-optimal, so not performing well on those models won’t affect my rating. Just curious how this method performs on LMMs.
>
> A2: We sincerely appreciate your insightful question and kind understanding regarding the detection performance. We are glad to explore more facets of our method following interesting comments and instructions. Following your valuable comments, we evaluate the detection performance of WePe using different multi-modal  models, such as BLIP (ViT-L/14) and CLIP (ViT-L/14), and list the results in the following table.
> |        | AUROC | AP    |
> |--------|-------|-------|
> | BLIP   | 68.25 | 64.68 |
> | CLIP   | 77.89 | 77.90 |
> | DINOv2 | 87.99 | 85.04 |
>
> The results show that these multimodal models do not perform as well as DINOv2, which may be due to the fact that the image features of the multimodal models are more focused on semantic information.
>
> >Q3: Could the author show the method’s scaling performance with increased n? If the model's predictive performance improves as the number of samples n increases, this would indirectly prove the unbiased nature of the method proposed in the article in terms of predictive uncertainty.
>
> A3: We sincerely appreciate your unbiased-estimation perspective! We agree that increasing the number of sample n would lead to unbiased estimation of uncertainty and improved detection performance. Fortunately, this is consistent with our experimental results shown Figure 7 (Appendix).
>
> Following your valuable suggestion, we further increase n from 20 to 30. The results are as follows. We can see that multiple perturbations can further improve the performance, but this comes with a higher computational cost. Moreover, the increasing performance proves the unbiased nature of our method. We have added the results and discussions to the revision.
> |      | AUROC | AP    |
> |------|-------|-------|
> | n=1  | 87.99 | 85.04 |
> | n=3  | 88.14 | 85.32 |
> | n=5  | 88.71 | 85.90 |
> | n=10 | 88.99 | 86.68 |
> | n=20 | 89.12 | 86.71 |
> | n=30 | 89.18 | 86.83 |

---

> ### Author Response · Authors · 2024-11-22
>
> >Q4: Since the key technical contribution of this method is weight perturbation, I’m interested to see how this method performs on adversarial examples? The author can get the adversarial examples by simply applying noises to the test images.
>
> A4: Thanks for the interesting insights! Following your suggestion, We add Gaussian noise to the natural image, the generated image, and the natural and generated images ,respectively. The results are given in the following additional table 1. We can see that when noise is injected, the feature similarity between the clean model and the noisy model for the noisy image decreases, which leads to a change in the detection performance. Aligning with your insightful comments in “Q3”, we find that increasing the number of samples can slightly mitigate the effect of adversarial examples, as shown in the following additional table 2.
>
> Additional table 1: Impact of noise on detection performance
>
> |                                                                | AUROC | AP    |
> |----------------------------------------------------------------|-------|-------|
> | Results on clean images                                        | 87.99 | 85.04 |
> | Adding noise to natural images(with standard deviation 0.05)   | 81.79 | 78.99 |
> | Adding noise to natural images(with standard deviation 0.1)    | 76.16 | 73.16 |
> | Adding noise to natural images(with standard deviation 0.15)   | 69.53 | 66.83 |
> | Adding noise to generated images(with standard deviation 0.05) | 88.03 | 85.82 |
> | Adding noise to generated images(with standard deviation 0.1)  | 88.77 | 87.14 |
> | Adding noise to genreated images(with standard deviation 0.15) | 90.25 | 88.99 |
> | Adding noise to all images(with standard deviation 0.05)       | 84.94 | 81.98 |
> | Adding noise to all images(with standard deviation 0.1)        | 81.46 | 78.25 |
> | Adding noise to al images(with standard deviation 0.15)        | 78.19 | 75.32 |
>
> Additional table 2: Reducing the impact of noise through ensemble
>
> |                                                                     | AUROC | AP    |
> |---------------------------------------------------------------------|-------|-------|
> | Results on clean images                                             | 87.99 | 85.04 |
> | n = 1, Adding noise to natural images(with standard deviation 0.05) | 81.79 | 78.99 |
> | n = 5, Adding noise to natural images(with standard deviation 0.05) | 85.30 | 82.59 |
> | n = 1, Adding noise to natural images(with standard deviation 0.1)  | 76.16 | 73.16 |
> | n = 5, Adding noise to natural images(with standard deviation 0.1)  | 81.36 | 78.39 |
> | n = 1, Adding noise to natural images(with standard deviation 0.15) | 69.53 | 66.83 |
> | n = 5, Adding noise to natural images(with standard deviation 0.15) | 75.16 | 72.58 |
>
>
> Reference:
>
> [r1] Gradient Cuff: Detecting Jailbreak Attacks on Large Language Models by Exploring Refusal Loss Landscapes. Xiaomeng Hu, Pin-Yu Chen, Tsung-Yi Ho

---

> ### Author Response · Authors · 2024-11-25
>
> Dear Reviewer #fpAT,
>
> Thank you very much for your time and valuable feedback.
>
> Here is a summary of our response for your convenience:
>
> **(1) Issues of comparison with Gradient Cuff:** We have compared Gradient Cuff to WePe and found that it further enhances WePe's performance.
>
> **(2) Issues in the performance of WePe on multimodal models:** We have implemented WePe on CLIP and BLIP, respectively. Experimental results show that WePe does not perform as well on these multimodal models as it does on DINOv2, which aligns with our previous discussions.
>
> **(3) Issues of scaling performance with increased n:** We have conducted relevant experiments showing the scaling performance with increased n.
>
> **(4) Issues of adversarial examples:** We have conducted experiments showing that adding noise causes samples to deviate from the original distribution, leading to changes in detection performance. This effect can be mitigated by increasing n.
>
> We understand you're busy. But as the window for responding and paper revision is closing, would you mind checking our response and confirm whether you have any further questions? We are very glad to provide answers and revision to your further questions.
>
> Best regards and thanks,
>
> Authors of #13682

---

> ### Author Response · Authors · 2024-11-27
>
> Dear Reviewer #fpAT,
>
> Thanks a lot for your time in reviewing and insightful comments, according to which we have carefully revised the paper to answer the questions. We sincerely understand you’re busy. However, as the paper revision window is about to close, would you mind checking the response and revision to confirm where you have any further questions?
>
> We are looking forward to your reply and happy to answer your further questions.
>
> Best regards
>
> Authors of #13682

---

> ### Author Response · Authors · 2024-11-29
>
> Dear Reviewer #fpAT,
>
> Thanks a lot for your time in reviewing and insightful comments. We sincerely understand you're busy. But as the window for discussion is closing, could you please take some time to review our responses? We'd love to get your further feedback and do our best to address it to enhance our work.
>
> We are looking forward to your reply!
>
> Best regards,
>
> Authors of #13682

---

> ### Author Response · Authors · 2024-11-30
> **Welcome for more discussions**
>
> Dear Reviewer #fpAT,
>
> Thanks a lot for your time in reviewing and insightful comments. We sincerely understand you're busy. But as the window for discussion is closing, could you please take some time to review our responses? We'd love to get your further feedback and do our best to address it to enhance our work.
>
> We are looking forward to your reply!
>
> Best regards,
>
> Authors of #13682

---

> ### Author Response · Authors · 2024-12-01
>
> Dear Reviewer #fpAT,
>
> Thanks a lot for your time in reviewing and insightful comments. We sincerely understand you're busy. But as the window for discussion is closing, could you please take some time to review our responses? We'd love to get your further feedback and do our best to address it to enhance our work.
>
> We are looking forward to your reply!
>
> Best regards,
>
> Authors of #13682

---

> ### Author Response · Authors · 2024-12-03
>
> Dear Reviewer fpAT,
>
> We really appreciate your constructive comments. Could you please check our response and confirm if your issues have been resolved? If so could you adjust your rating as you said?

---

> ### Author Response · Authors · 2024-12-04
>
> Dear Reviewer fpAT,
>
> Could you please check our response and confirm if your issues have been resolved?

---

### Official Review · Reviewer_ANn9 · 2024-11-06

**Soundness:** 3
**Presentation:** 4
**Contribution:** 2
**Rating:** 6
**Confidence:** 3

**Summary:**

The authors use pre-trained DNN image models to generate an uncertainty prediction regarding whether an input image is synthetic or real.  The method involves perturbing the model weights and observing a change in the features extracted from an image.  The results suggest that synthetic data features are more affected by model weight perturbations than real data features.  This is demonstrated using three benchmark datasets and DINOv2.

**Strengths:**

The paper is well written and easy to follow.  The methodology is intuitive, and is implementable beyond the studies provided by the authors.  Distinguishing between real and synthetic data is an open question in the community.

**Weaknesses:**

The method present in this paper exclusively relies on access to deep learning models which have not been trained on any generated synthetic data.  Unfortunately, the proliferation of synthetic images means that such models will become harder and harder to find as this area of research progresses.  The method may have a built-in "expiration date", in that, future state-of-the-art models will likely be tainted (either knowingly or unknowingly) with generated data.  Similarly, as synthetic data becomes closer to natural data, I would expect the synthetic features to also approximate natural features.

The authors directly acknowledge there is no theoretical justification for this method, and list it under future work. I appreciate their honesty and clarity, and agree that the result is very interesting.  However, without a theoretical justification or a more extensive analysis of the differences in natural-synthetic feature representation driving this metric, this work is unlikely to become high impact.

**Questions:**

None

---

> ### Author Response · Authors · 2024-11-22
>
> We sincerely appreciate your dedicated time reviewing our paper and are grateful for your constructive comments. According to your constructive comments, we provide detailed feedback below, which has been added into our revision. We hope our feedback can address your concerns and improve our work.
>
> >Question 1: The method present in this paper exclusively relies on access to deep learning models which have not been trained on any generated synthetic data. Unfortunately, the proliferation of synthetic images means that such models will become harder and harder to find as this area of research progresses. The method may have a built-in "expiration date", in that, future state-of-the-art models will likely be tainted (either knowingly or unknowingly) with generated data.
>
> Answer 1: Thank you for your insightful and pragmatic perspective. We deeply agree that, with the proliferation of generated images, future large-scale models could indeed become contaminated by such data, making it increasingly difficult to distinguish between natural and generated images.
> Your valuable insight highlights a crucial point regarding the mitigation of this contamination issue. One potential solution is to employ machine unlearning [r1] techniques to detect and address generated images. Machine unlearning focuses on removing the influence of specific data from pre-trained models, primarily due to privacy concerns. In this context, when generative images are incorporated into the training process of large-scale models, we can utilize machine unlearning techniques to eliminate the effects of these images on pre-trained models. This approach would help ensure that the features of natural and generated images remain distinct and separable.
>
> > Question 2: As synthetic data becomes closer to natural data, I would expect the synthetic features to also approximate natural features.
>
> Answer 2: Thank you for your in-depth perspective. As generative models continue to advance, the distinction between generated and natural images is becoming increasingly subtle. When synthetic images become indistinguishable from natural ones, it will indeed pose significant challenges in differentiating between them. This potential development underscores the urgent need to create robust detection methods. Developing these methods is essential to address the potential misuse of generated images and to ensure the integrity of visual data. We are committed to advancing detection technologies to meet these emerging challenges and hope to get your valuable support.
>
> > Question 3: The authors directly acknowledge there is no theoretical justification for this method, and list it under future work. I appreciate their honesty and clarity, and agree that the result is very interesting. However, without a theoretical justification or a more extensive analysis of the differences in natural-synthetic feature representation driving this metric, this work is unlikely to become high impact.
>
> Answer 3: We appreciate your thoughtful feedback and concerns. The hypothesis regarding the distribution discrepancy between natural and generated images is foundational to our work; without it, distinguishing between the two would be impossible. Our experiments have empirically observed this discrepancy, not only in DINOv2 but also in the representation space of CLIP. We will include these additional results in the revised version of our paper.
>
> Furthermore, numerous outstanding works have utilized uncertainty approaches to detect distribution discrepancies, providing a basis for our ongoing efforts. We aim to theoretically substantiate the existence of the described distribution discrepancy through the learning processes of diffusion models. However, we acknowledge that achieving this within a single study is a significant challenge. Therefore, we plan to refine and expand upon this aspect in our future research.
>
> Reference:
>
> [r1]: Large language model unlearning

---

> ### Author Response · Authors · 2024-11-25
>
> Dear Reviewer #ANn9,
>
> Thank you very much for your time and valuable feedback.
>
> Here is a summary of our response for your convenience:
>
> **(1) Issues of model contamination by generated images:** With the proliferation of generated images, it is likely that future models will be contaminated by these images. However, techniques like machine unlearning may help mitigate the effects of generated images on models.
>
> **(2) Issues of synthetic data getting closer to natural data:** The development of generative models has led to increasingly subtle differences between generated and natural images. This highlights the importance of developing robust detection methods, which we are actively working on.
>
> **(3) Issues about theoretical justification:** The hypothesis regarding the distribution discrepancy between natural and generated images is foundational to our work; without it, distinguishing between the two would be impossible. We observe this discrepancy in both DINOv2 and CLIP representation spaces. We will attempt to theoretically substantiate the existence of this distribution discrepancy through the learning processes of diffusion models.
>
> We understand you're busy. But as the window for responding and paper revision is closing, would you mind checking our response and confirm whether you have any further questions? We are very glad to provide answers and revision to your further questions.
>
> Best regards and thanks,
>
> Authors of #13682

---

> > ### Comment · Reviewer_ANn9 · 2024-11-25
> >
> > I thank the authors for their response to my comments and to those of the other reviewers.  The work is interesting, but I feel that the contributed value continues to rely too heavily on future work.  Unfortunately, I am not able to improve my score.
> >
> > 1) The claim that unlearning would be effective for re-creating the differences in the representation space of the data is not demonstrated in your paper. You have not presented results for your experiments which show that an unlearning technique would correctly map the contaminated data back to its original representation.
> >
> > 2) I agree that robust methods are required to address this issue.  You state that you are "currently working on this".  I encourage you to continue this work, and it would be my privilege to review such work if given the opportunity.  However, this is not demonstrated in the current paper.
> >
> > 3) I agree that this hypothesis is central to your work.  However, my concern is that your hypothesis as currently presented will be nullified, and you have not presented a justification for why this would not be the case.  To my knowledge, no such rule or theory for why synthetic data will continue to be differentiable from real data in the representation space *does* exist.  If the authors can present peer reviewed work presenting support for such an argument, I will reconsider my score.

---

> > > ### Author Response · Authors · 2024-11-26
> > >
> > > Thanks for your prompt reply!
> > >
> > > >Q1: The claim that unlearning would be effective for re-creating the differences in the representation space of the data is not demonstrated in your paper. You have not presented results for your experiments which show that an unlearning technique would correctly map the contaminated data back to its original representation.
> > >
> > > A1: Thank you for your comment. We acknowledge that we are currently unable to verify the effectiveness of this method. At this stage, we have not found a large vision model that has been trained using generated images, and we are not in a position to independently train such a model in a short period of time. However, we agree that this is a valid concern, and the direction of machine unlearning might help address it. To provide an initial indication of feasibility, we conducted a simple experiment. We trained a triple classifier using images of natural cat, natural dog, natural bird, and generated bird, where natural and generated birds were grouped together. After training, we performed unlearning finetuning, applying gradient descent on the natural images and gradient ascent on the generated bird images. The results (classification accuracy) are shown below. The unlearning process successfully causes the model to "forget" the generated bird.
> > >
> > > |                   | natural cat | natural dog | natural bird | generated bird |
> > > |-------------------|-------------|-------------|--------------|----------------|
> > > | before unlearning | 97.74       | 96.95       | 98.43        | 97.85          |
> > > | after unlearning  | 96.76       | 95.89       | 96.87        | 6.48          |
> > >
> > > >Q2: I agree that robust methods are required to address this issue. You state that you are "currently working on this". I encourage you to continue this work, and it would be my privilege to review such work if given the opportunity. However, this is not demonstrated in the current paper.
> > >
> > > A2: Thank you for your comments. We deeply agree with you. As synthetic data becomes increasingly similar to natural data, and synthetic features more closely resemble natural features, it becomes progressively more difficult to distinguish between the two. As generative models continue to evolve, current detection techniques may gradually become less effective. However, for the time being, our approach achieves the best average performance.
> > >
> > > >Q3: I agree that this hypothesis is central to your work. However, my concern is that your hypothesis as currently presented will be nullified, and you have not presented a justification for why this would not be the case. To my knowledge, no such rule or theory for why synthetic data will continue to be differentiable from real data in the representation space does exist. If the authors can present peer reviewed work presenting support for such an argument, I will reconsider my score.
> > >
> > > A3: Thank you for your comment. We would like to clarify this further. The core assumption we make is that there is **data distribution discrepancy** between natural and generated images. This assumption is valid for current generative models and has been confirmed by many works[ref1, ref2, ref3]. This assumption is also the foundation of many generative image detection methods (we cannot distinguish between images that are indistinguishable).
> > >
> > > Secondly, we observe that this discrepancy in data distribution can be captured by the representation space of a vision model pre-trained on a large number of natural images, i.e., there is **feature distribution discrepancy** between the generated and natural images. However, this remains an observation, and we have not found theoretical proof despite reviewing the literature. We only observe a similar phenomenon in UnivFD [ref4], where the feature distribution discrepancy is observed in the representation space of CLIP:ViT-L/14.
> > >
> > > That said, we can confirm the existence of feature distribution discrepancy of generated and natural images based on an important metric for evaluating generative models, the **FID score**. The FID score measures the feature distribution discrepancy between natural and generated images on the Inception network. When the FID score is 0, it indicates that the two distributions do not differ. However, even on these simple networks such as Inception v3, advanced generative models like ADM still achieve an FID score of 11.84, not to mention that on powerful models such as DINOv2, we observe significant feature distribution discrepancy.
> > >
> > > Reference:
> > >
> > > [ref1]: Intriguing properties of synthetic images: from generative adversarial networks to diffusion models, CVPR2023
> > >
> > > [ref2]: Rethinking the up-sampling operations in cnn-based generative network for generalizable deepfake detection, CVPR2024
> > >
> > > [ref3]: AEROBLADE: Training-Free Detection of Latent Diffusion Images Using Autoencoder Reconstruction Error, CVPR2024
> > >
> > > [ref4]: Towards universal fake image detectors that generalize across generative models, CVPR2023

---

> > > ### Author Response · Authors · 2024-11-27
> > >
> > > Dear Reviewer #ANn9,
> > >
> > > Thank you so much for your profound insights. Your suggestions have further motivated us to use FID scores to measure the feature distribution discrepancy between natural and generated images. To avoid the effects of categories, we compute the FID scores using the DINOv2 model on the LSUN-BEDROOM benchmark. For each category of images, we randomly select 5000 images for calculation.
> > >
> > > In addition to calculating the FID scores between natural images and generated images, we also calculate the FID scores between natural images and natural images. As shown in the table below, the FID scores between natural images and generated images are significantly higher than the FID scores between natural images and natural images. Moreover, there is a clear positive correlation between the detection performance of WePe and the FID score. This result fully explains the existence of feature distribution discrepancy between natural and generated images on DINOv2, and demonstrates that WePe can effectively detect the feature distribution discrepancy.
> > >
> > > Thanks again for your constructive feedback. Your insights greatly enhance our work. We look forward to your reply and are happy to answer any further questions.
> > >
> > > | Models    | Natural | ADM   | StyleGAN | iDDPM | DDPM  | Diffusion GAN | Unleashing Transformer | Projected GAN  |
> > > |-----------|---------|-------|-----------|-------|-------|---------------|----------------|------------|
> > > | FID score | 1.09    | 18.25 | 52.31     | 59.94 | 80.44 | 116.06        |     130.00    |    132.75   |
> > > | AUROC     | 50.00   | 73.85 | 83.50     | 86.23 | 88.84 | 94.16         |      94.18     |    95.34   |
> > >
> > > Best regards and thanks,
> > > Authors of #13682

---

> ### Author Response · Authors · 2024-11-29
>
> Dear Reviewer #ANn9,
>
> Thanks a lot for your time in reviewing and insightful comments. We sincerely understand you're busy. But as the window for discussion is closing, could you please take some time to review our responses? We'd love to get your further feedback and do our best to address it to enhance our work.
>
> We are looking forward to your reply!
>
> Best regards,
>
> Authors of #13682

---

> ### Author Response · Authors · 2024-11-30
> **Welcome for more discussions**
>
> Dear Reviewer #ANn9,
>
> Thanks a lot for your time in reviewing and insightful comments. We sincerely understand you're busy. But as the window for discussion is closing, could you please take some time to review our responses? We'd love to get your further feedback and do our best to address it to enhance our work.
>
> We are looking forward to your reply!
>
> Best regards,
>
> Authors of #13682

---

> ### Author Response · Authors · 2024-12-01
>
> Dear Reviewer #ANn9,
>
> Thanks a lot for your time in reviewing and insightful comments. We sincerely understand you're busy. But as the window for discussion is closing, could you please take some time to review our responses? We'd love to get your further feedback and do our best to address it to enhance our work.
>
> We are looking forward to your reply!
>
> Best regards,
>
> Authors of #13682

---

> ### Author Response · Authors · 2024-12-03
> **Window for discussion is closing**
>
> Dear Reviewer ANn9,
>
> Thanks a lot for your time in reviewing and insightful comments. We sincerely understand you're busy. But as the window for discussion is closing, could you please check to see if your concerns have been addressed? We'd love to get your further feedback and do our best to address it to enhance our work.

---

### Author Response · Authors · 2024-12-02

Dear reviewers,

Could you please check our responses?

---

### Meta-Review · Area_Chair_1PkA · 2024-12-18

**Metareview:**

The recommendation is based on the reviewers' comments, the area chair's evaluation, and the author-reviewer discussion.

While the reviewers see some merits in using the proposed uncertainty metric to differentiate AI-generated and natural images, this submission should not be accepted in its current form due to several fundamental issues, as pointed out by the reviewers, including

- Unconvincing justification of the difference between this work and existing works. Note that the concerns on computation cost have been resolved. The AC also certifies that Reviewer JnK7 is not an author of the mentioned existing work, RIGID. The reviewer does not have a direct conflict of interest in asking the authors about the questions related to RIGID.
- Insufficient justification of novelty: Reviewers pointed out that (1) a similar "weight perturbation" method was previously explored in WeiPer [R1] for OOD detection, and (2) the approximate uncertainty used in the proposed method is essentially equivalent to a simple cosine similarity (as was used in RIGID [R2]).
- The work could be improved by providing some theoretical justification. Reviewers pointed out that the concern of unbiased estimator remains unresolved.
- Experiments need further evaluation and analysis: For example, in the GenImage task, WePe underperformed one baseline in several models. The authors did not elaborate on this while claiming WePe outperformed the baselines in all tasks based on the average performance.
- Lack of motivation: as the FID score only performs well in very narrow cases and is no longer used for training generative AI models for that reason, without a more rigorous justification as to why it is reasonable to expect that there will continue to be a distribution gap between real and synthetic data (especially given the ongoing efforts to minimize the current gap), the underlying working mechanism remains unclear.

[R1] Granz, Maximilian, et al. ‘WeiPer: OOD Detection Using Weight Perturbations of Class Projections’. The Thirty-Eighth Annual Conference on Neural Information Processing Systems, 2024, https://openreview.net/forum?id=8HeUvbImKT.

[R2] He, Zhiyuan, Pin-Yu Chen, and Tsung-Yi Ho. "RIGID: A Training-free and Model-Agnostic Framework for Robust AI-Generated Image Detection." arXiv preprint arXiv:2405.20112 (2024).

During the final discussion phase, most reviewers agreed to reject this submission, due to the primary concerns listed above and in the reviewers' responses. Overall, this paper requires significant modifications and another round of full review. I hope the reviewers’ comments can help the authors prepare a better version of this submission.

**Additional Comments On Reviewer Discussion:**

This submission should not be accepted in its current form due to several fundamental issues, as pointed out by the reviewers, including

- Unconvincing justification of the difference between this work and existing works. Note that the concerns on computation cost have been resolved. The AC also certifies that Reviewer JnK7 is not an author of the mentioned existing work, RIGID. The reviewer does not have a direct conflict of interest in asking the authors about the questions related to RIGID.
- Insufficient justification of novelty: Reviewers pointed out that (1) a similar "weight perturbation" method was previously explored in WeiPer [R1] for OOD detection, and (2) the approximate uncertainty used in the proposed method is essentially equivalent to a simple cosine similarity (as was used in RIGID [R2]).
- The work could be improved by providing some theoretical justification. Reviewers pointed out that the concern of unbiased estimator remains unresolved.
- Experiments need further evaluation and analysis: For example, in the GenImage task, WePe underperformed one baseline in several models. The authors did not elaborate on this while claiming WePe outperformed the baselines in all tasks based on the average performance.
- Lack of motivation: as the FID score only performs well in very narrow cases and is no longer used for training generative AI models for that reason, without a more rigorous justification as to why it is reasonable to expect that there will continue to be a distribution gap between real and synthetic data (especially given the ongoing efforts to minimize the current gap), the underlying working mechanism remains unclear.

[R1] Granz, Maximilian, et al. ‘WeiPer: OOD Detection Using Weight Perturbations of Class Projections’. The Thirty-Eighth Annual Conference on Neural Information Processing Systems, 2024, https://openreview.net/forum?id=8HeUvbImKT.

[R2] He, Zhiyuan, Pin-Yu Chen, and Tsung-Yi Ho. "RIGID: A Training-free and Model-Agnostic Framework for Robust AI-Generated Image Detection." arXiv preprint arXiv:2405.20112 (2024).

During the final discussion phase, most reviewers agreed to reject this submission, due to the primary concerns listed above and in the reviewers' responses.

---

### Decision · Program_Chairs · 2025-01-22

Reject